# Distinct trafficking routes of polarized and non-polarized membrane cargoes in *Aspergillus nidulans*

**Georgia Maria Sagia[1], Xenia Georgiou[1], Georgios Chamilos[2,3], George Diallinas[1,2]\*, Sofia Dimou[1]**

[1]Department of Biology, National and Kapodistrian University of Athens, Panepistimioupolis, Athens, Greece; [2]Institute of Molecular Biology and Biotechnology, Foundation for Research and Technology, Heraklion, Greece; [3]School of Medicine, University of Crete, Heraklion, Greece

**Abstract** Membrane proteins are sorted to the plasma membrane via Golgi-dependent trafficking. However, our recent studies challenged the essentiality of Golgi in the biogenesis of specific transporters. Here, we investigate the trafficking mechanisms of membrane proteins by following the localization of the polarized R-SNARE SynA versus the non-polarized transporter UapA, synchronously co-expressed in wild-type or isogenic genetic backgrounds repressible for conventional cargo secretion. In wild-type, the two cargoes dynamically label distinct secretory compartments, highlighted by the finding that, unlike SynA, UapA does not colocalize with the late-Golgi. In line with early partitioning into distinct secretory carriers, the two cargoes collapse in distinct ER-Exit Sites (ERES) in a *sec31*[ts] background. Trafficking via distinct cargo-specific carriers is further supported by showing that repression of proteins essential for conventional cargo secretion does not affect UapA trafficking, while blocking SynA secretion. Overall, this work establishes the existence of distinct, cargo-dependent, trafficking mechanisms, initiating at ERES and being differentially dependent on Golgi and SNARE interactions.

**\*For correspondence:**
diallina@biol.uoa.gr

**Competing interest:** The authors declare that no competing interests exist.

## Editor's evaluation

This fundamental study advances our understanding of an unconventional route by which certain transmembrane proteins reach the plasma membrane of the fungus Aspergillus nidulans. By carefully examining two model plasma membrane proteins in parallel, the authors provide compelling evidence for their earlier proposal that sorting of the two proteins diverges upon ER export, with one protein following the standard secretory pathway while the other protein follows a Golgi-independent route. Even though a mechanistic understanding of the new pathway is not yet available, this work will be of interest to cell biologists who study membrane traffic.

## Introduction

In eukaryotic cells proper subcellular trafficking, topological distribution and targeting of de novo made plasma membrane (PM) proteins constitute processes of primary importance (*Barlowe and Helenius, 2016*). The great majority of these proteins translocate co-translationally from ribosomes into the lipid bilayer of the endoplasmic reticulum (ER) (*Voorhees and Hegde, 2016*). Proper integration into the ER membrane is followed by partitioning of membrane cargoes into ER-Exit Sites (ERES), which define distinct microdomains promoting the bud-off of vesicles (*Zanetti et al., 2013*; *D'Arcangelo et al., 2013*; *Feyder et al., 2015*; *Gomez-Navarro and Miller, 2016*) or tubules (*Shomron*

*et al., 2021*; *Weigel et al., 2021*; *Raote and Malhotra, 2021*; *Phuyal and Farhan, 2021*). Vesicles or tubules loaded with membrane cargoes are directed to and fuse with the early- or *cis*-Golgi, seemingly via a dynamic pre-Golgi compartment, known as ERGIC (ER-to-Golgi Intermediate Compartment) in mammals and yeast, or the Golgi entry core compartment (GECCO) in plant cells (*Aridor, 2018*; *Peotter et al., 2019*; *Wong-Dilworth et al., 2023*; *Fougère et al., 2023*; *Tojima et al., 2023*). Once in the early-Golgi, cargoes 'advance' toward the medial- and late- or trans-Golgi network (TGN), through Golgi maturation (*Glick and Luini, 2011*; *Suda and Nakano, 2012*; *Day et al., 2013*; *Pantazopoulou and Glick, 2019*; *Lujan and Campelo, 2021*; *Tojima et al., 2023*). From the late-Golgi/TGN, cargoes destined to the PM are loaded in vesicular carriers coated, in most cases, with the AP-1/clathrin complex (*Robinson, 2015*; *Zeng et al., 2017*; *Casler et al., 2019*). Alternative coated vesicles for specific cargoes or proteins other than those destined to the PM are also well studied (*Makowski et al., 2020*). The movement of such vesicular carriers toward the PM can be direct or via specific sorting endosomes and seems to require functional microtubule tracks and actin filaments (*Berepiki et al., 2011*; *Schultzhaus et al., 2016*; *Fourriere et al., 2020*).

The initial exit of PM cargoes from ERES is orchestrated primarily by the protein coat complex COPII (Sar1, Sec23/24, Sec13/Sec31) and its associated scaffold proteins or regulators (e.g., Sec12, Sec16, cargo adaptors, or chaperones) (*Zanetti et al., 2013*; *Miller and Schekman, 2013*; *Peotter et al., 2019*). Past and recent results have also strongly supported the direct involvement of coat complex COPI (*Shomron et al., 2021*; *Weigel et al., 2021*; *Raote and Malhotra, 2021*; *Phuyal and Farhan, 2021*; *Wong-Dilworth et al., 2023*), independently to its long-thought role in specific cargo retrograde trafficking from the Golgi to the ER (*Altan-Bonnet et al., 2004*; *Gomez-Navarro and Miller, 2016*). Based on recent findings, COPII and COPI seem to carry topologically distinct functions, in which COPII first concentrates and drives cargoes into ERES, and subsequently COPI promotes further extension of budding, leading to homotypic or heterotypic fusion of the vesicular–tubular carriers into specific pre-Golgi intermediates, such as the ERGIC. Notably, both COPII and COPI protein complexes polymerize into flexible curved scaffolds to support membrane deformation, which seems crucial for cargo trafficking at the ER to Golgi interface. Anterograde cargo trafficking also necessitates membrane fusion at least in two steps. First, at the ER-to-Golgi transfer and secondly at the targeting of post-Golgi vesicles or endosomal carriers to the PM. Membrane fusion between cargo carriers and recipient membranes requires specific tethering factors, SNARE proteins (soluble *N*-ethylmaleimide-sensitive factor attachment protein receptors) and Rab GTPases, all of which regulate organelle identity as well as the direction of vesicular transport (*Cai et al., 2007*; *Stenmark, 2009*; *Novick, 2016*; *Stanton and Hughson, 2023*).

The outlined mechanism of anterograde cargo trafficking, which is centrally dependent on Golgi maturation and functioning, is thought to be the major mechanism that directs membrane cargoes to the PM, excluding specific cases of unconventional trafficking routes, detected mostly under cellular stress and only for a small number of specific cargoes (*Rabouille, 2017*; *Gee et al., 2018*; *Camus et al., 2020*; *Zhang et al., 2020*; *Kemal et al., 2022*; *Sun et al., 2024*). However, our view of canonical anterograde cargo trafficking has been mostly shaped by studies in a small number of cell systems, such as yeasts or specific mammalian or plant cells, or by in vitro biochemical approaches, and crucially, with only a limited number of membrane cargoes. Surprisingly, trafficking mechanisms of the two most abundant types of membrane cargoes in eukaryotic cells, namely transporters and receptors, have not been investigated systematically. Related to this issue, we have recently provided evidence that in the filamentous fungus *Aspergillus nidulans* several nutrient transporters, the major proton pump ATPase PmaA and the PalI component of the pH sensing receptor, can all well translocate to the PM when key Golgi proteins are repressed (*Dimou and Diallinas, 2020*; *Dimou et al., 2022*; *Dimou et al., 2020*). This contrasts what has been observed, using the same fungal system, with apical PM proteins, such as chitin synthase ChsB$^{Chs1/2}$, R-SNARE SynA$^{Snc1/2}$, or lipid flippases DnfA/B$^{Dnf1/2}$, which are considered as standard Golgi-dependent cargoes (*Hernández-González et al., 2018*; *Taheri-Talesh et al., 2008*; *Pantazopoulou and Peñalva, 2011*; *Schultzhaus et al., 2015*; *Schultzhaus et al., 2016*; *Martzoukou et al., 2018*; *Dimou et al., 2020*). A notable aspect of our previous work is also that both Golgi-dependent and Golgi-independent cargo trafficking seems to depend on key COPII components. This in turn signified that there must be two subpopulations of COPII cargo carriers: one that reaches the Golgi carrying conventional, growth-related, apically localized cargoes, and a second one that bypasses the Golgi to sort non-polarized cargoes needed for nutrition or homeostasis to the PM.

Here, we describe a novel genetic system in *A. nidulans* for synchronously co-expressing and following the de novo localization of the non-polarized cargo UapA, a well-studied purine transporter, versus the polarly localized R-SNARE SynA. Using this system, we obtained multiple evidence for the existence of distinct cargo trafficking routes for these two cargoes, initiating at ERES and early secretory compartments. Most interestingly, we also obtained evidence that translocation of UapA to the PM seems to necessitate non-canonical SNARE interactions and is also exocyst-independent. Our work provides strong evidence that in *A. nidulans* specific polarized and non-polarized PM cargoes serving different physiological functions follow distinct anterograde trafficking routes.

## Results

### Synchronously co-expressed polarized (SynA) and non-polarized (UapA) membrane cargoes follow distinct routes to the PM

A new genetic system was designed to follow the dynamic subcellular localization of selected specific polarized and non-polarized membrane cargoes, namely SynA$^{Snc1/2}$ (R-SNARE) versus UapA (purine transporter), co-expressed in parallel. Notice that in this work we keep the *A. nidulans* protein (or gene) terminology based on previous experimental work and relevant publications, albeit in cases of proteins not studied before and/or not being annotated, we mostly use the terminology of the orthologous *Saccharomyces cerevisiae* proteins/genes. When using the *A. nidulans* terminology of proteins, we also give the name of the orthologous protein in superscript, when these appear in the text for the first time (all genes/proteins with their *A. nidulans* annotation numbers used in this study are shown in *Supplementary file 1*). SynA and UapA tagged with GFP have been used previously for studying cargo trafficking mechanisms, albeit expressed in different strains, and in some cases under different promoters. In these studies, SynA has been shown to localize polarly in the apex of growing hyphae via conventional Golgi and post-Golgi secretion (*Taheri-Talesh et al., 2008*; *Martzoukou et al., 2018*), whereas UapA localized non-polarly all over the hyphal PM, through a mechanism that essentially did not require Golgi maturation and post-Golgi secretion (*Dimou et al., 2020*; *Dimou et al., 2022*).

Here, we constructed novel strains where we can simultaneously co-express, via the regulatable *alcA* promoter (*alcA$_p$*), functional mCherry-SynA and UapA-GFP. In these strains, the respective engineered DNA cassettes replaced the endogenous loci via standard *A. nidulans* reverse genetics. Transcription from the *alcA$_p$* can be repressed by glucose and derepressed in poor carbon sources (e.g., fructose). Derepression of *alcA$_p$* by fructose leads to medium levels of expression (*Flipphi et al., 2003*). *Figure 1A* shows that the strain co-expressing *alcA$_p$*-UapA-GFP and *alcA$_p$*-mCherry-SynA grows similar to an isogenic wild-type (wt) strain in the presence of uric acid as a N source (i.e., substrate of UapA) and fructose as a C source, as expected. On the other hand, on uric acid/glucose media the *alcA$_p$*-UapA-GFP/*alcA$_p$*-mCherry-SynA strain resembles the Δ*uapA* null mutant, confirming that *uapA* transcription is well repressed. Notice that lack of SynA expression, either in the null mutants or in the repressible *alcA$_p$*-mCherry-SynA strain, does not affect growth (see also later).

We followed the subcellular steady-state accumulation of de novo expressed UapA and SynA in germlings and young hyphae after 4 hr derepression, in the *alcA$_p$*-UapA-GFP/*alcA$_p$*-mCherry-SynA 'dual-cargo' strain by using widefield fluorescence microscopy (*Figure 1B*). UapA-GFP labeled homogeneously the PM in a non-polarized or even anti-polarized manner, while mCherry-SynA marked mostly the PM of the apical region of hyphae, with some overlap with UapA-GFP, at the subapical region. Thus, UapA-GFP and mCherry-SynA nicely reflect non-polarized and polarized cargo accumulation when expressed in the same cell. The distinct distribution of the two cargoes remained the same at all development stages tested (e.g., germlings, young or mature hyphae). Noticeably however, the localization of UapA in some cells where SynA has not yet accumulated at the apical region, can also show apical localization. This points to the hypothesis that there is no specific inhibitory mechanism that excludes UapA from the apex but is rather the consequence of massive accumulation of apical markers related to fast growth of hyphae that limits translocation of UapA at the tips. This idea is strongly supported later, as in cases where we blocked conventional apical localization of SynA, UapA could then accumulate also in apical tips (see Figure 6C, E, F).

To investigate whether the observed steady-state accumulation of cargoes is the result of distinct or overlapping trafficking routes, especially at early steps of their secretion, we followed the dynamic

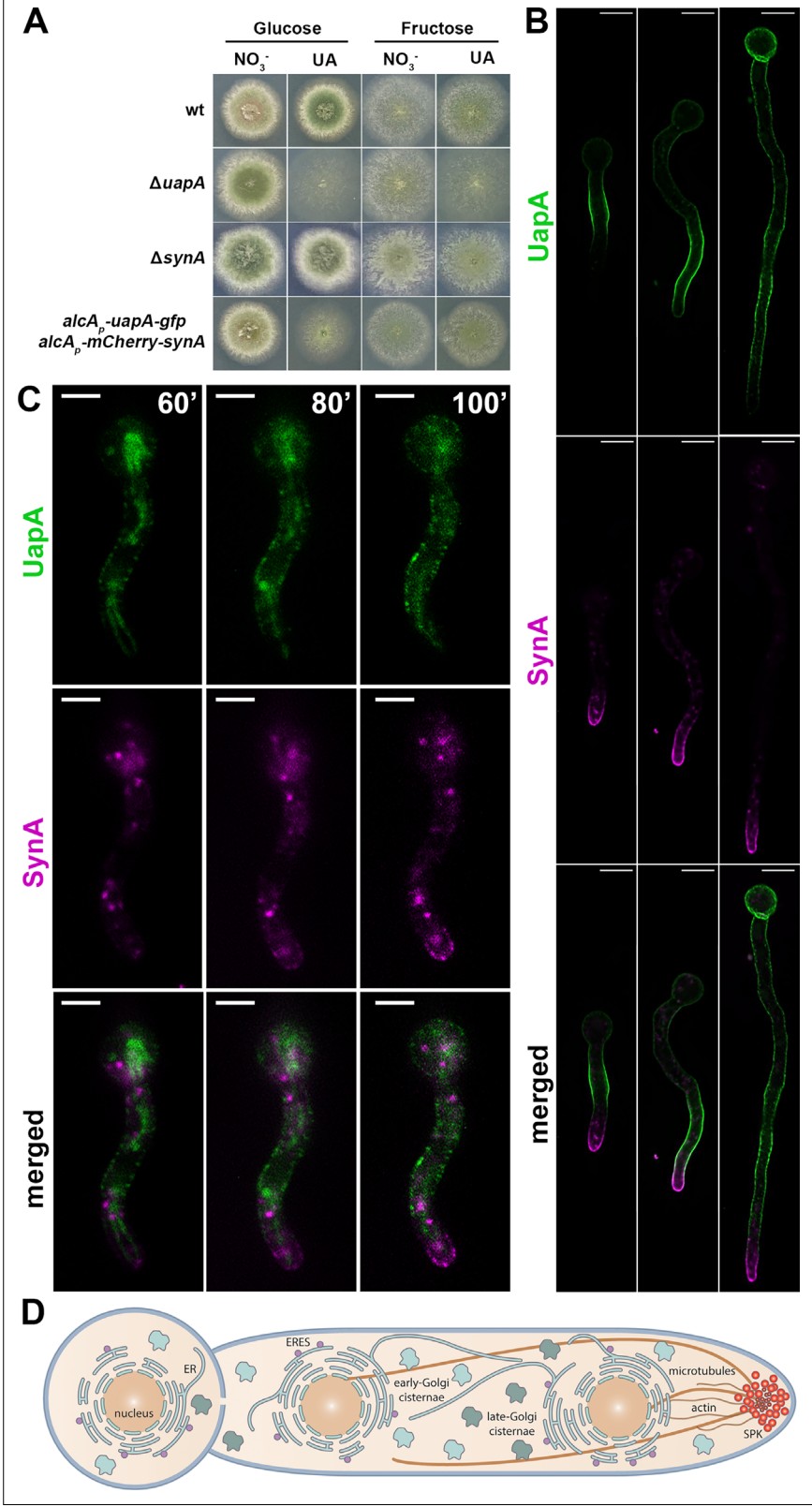

**Figure 1.** Synchronously co-expressed UapA and SynA follow distinct routes to the plasma membrane (PM) (widefield microscopy). (**A**) Growth test analysis of the strain co-expressing *alcA_p-uapA-gfp* and *alcA_p-mCherry-synA* compared to isogenic wt and Δ*uapA* or Δ*synA* strains. Notice that the *alcA_p-uapA-gfp* and *alcA_p-mCherry-synA* strain grows similar to the wt in the presence of uric acid as a N source (substrate of UapA) and fructose

*Figure 1 continued on next page*

*Figure 1 continued*

as a C source. On the other hand, on uric acid/glucose media the *alcA_p-uapA-gfp* and *alcA_p-mCherry-synA* strain resembles the Δ*uapA* mutant, confirming that *uapA* transcription is well repressed. Notice also that lack of SynA expression, either in the null mutants or in the repressible *alcA_p-mCherry-synA* strain, does not affect growth. (**B**) Maximal intensity projections of deconvolved snap shots showing steady-state localization of de novo expressed UapA and SynA in germlings and hyphae after 4–6 hr derepression. UapA-GFP labels homogeneously the PM in a non-polarized or even anti-polarized manner, while mCherry-SynA marks mostly the PM of the apical region of hyphae, with some overlap with UapA-GFP, at the subapical region. (**C**) Dynamic localization of de novo UapA/SynA after 1 hr derepression. On their way to the PM, while both cargoes are still in cytoplasmic structures, they label distinct compartments that do not seem to overlap significantly at any stage before their accumulation to the PM. (**D**) Schematic representation of an *A.nidulans* germling depicting the subcellular organization of key components of the secretory pathway (ER, ER-Exit Sites [ERES], early-, late-Golgi, cytoskeleton). Scale bars: 5 µm.

appearance of the two cargoes from the onset of their transcriptional derepression. Widefield epifluorescence microscopy showed that in most cells the two cargoes become visible after ~60 min of initiation of transcription and progressively labeled distinct cytoplasmic compartments, before accumulating in the PM. Importantly, the two cargoes did not seem to overlap significantly at any stage before accumulation to the PM (*Figure 1C*).

To better define the relative topology and dynamics of trafficking of SynA and UapA, we employed high-speed spinning-disc confocal microscopy and analysis of deconvolved z-stack images (i.e., dual-color 4D imaging; see Material and methods). *Figure 2A, B* and *Figure 2—videos 1–3* show snap shots and movies of UapA and SynA trafficking dynamics after synchronous derepression of transcription for 3 hr. The topology of UapA-labeled versus SynA-labeled structures was clearly distinct in all cells tested, as most UapA and SynA structures did not colocalize in movies recorded for a time period of 7 min (Pearson's correlation coefficient [PCC] = 0.01, *n* = 52). We detected only a small number of doubly labeled, mostly non-cortical puncta (0.33 ± 0.03 µm in diameter) (*Figure 2A*, see merged channels in middle and lower panels), suggesting that UapA and SynA might very partially colocalize on an early secretory compartment, before reaching the subapical or apical PM, respectively. Both UapA and SynA structures showed low-range mobility, but in addition several SynA puncta also moved toward the apical area (see *Figure 2—video 3* and *Figure 3—video 4*). In several hyphae, UapA appeared to label cytoplasmic oscillating thread structures decorated by pearl-like foci (0.32 ± 0.02 µm) as well as a very faint vesicular/tubular network (*Figure 2B* and associated movie in *Figure 2—video 2*). Similar thread-like structures with pearl-like foci were also seen with SynA (0.30 ± 0.03 µm). Larger SynA puncta were also observed (0.52 ± 0.08 µm) (*Figure 2—video 3*). Some UapA structures were tracked moving laterally toward the PM (indicated by yellow arrow heads in *Figure 2A, B*, see lower left panel). The average time needed for this to occur was estimated to be 12–18 s. The time frame in our videos ranged from 3 to 9 s, as shorter times proved too low for detecting the cargoes, especially SynA.

To identify whether the cytoplasmic structures labeled with UapA-GFP or mCherry-SynA, especially those that appear as small cytosolic puncta, reflect sorting via ERES and Golgi compartments, we performed colocalization studies with markers of ERGIC/early-Golgi (SedV^Sed5) or late-Golgi/TGN PH^OSBP-containing protein (*Pantazopoulou and Peñalva, 2009*; *Figure 3—videos 1–4*). Results shown in *Figure 3A, B* reveal that in movies recorded for a time period of 6–10 min UapA did not colocalize with any of the two Golgi markers [PCC = 0.01, *n* = 100 (UapA-SedV), PCC = 0.01, *n* = 35 (UapA-PH^OSBP)], while SynA colocalized significantly with the late-Golgi marker PH^OSBP (PCC = 0.74, *n* = 76, p < 0.0001), and very little with the ERGIC/early-Golgi marker SedV (PCC = 0.04, *n* = 40). Significant colocalization of SynA with the late-Golgi marker PH^OSBP is also confirmed via Super Resolution Radial Fluctuation (SRRF) imaging (*Figure 3—figure supplement 1*). Given that colocalization of SynA and PH^OSBP occurred all over the cytoplasm of hyphae and not only at the apical region, and because we record colocalization of cargoes before their steady-state accumulation to the PM, thus at a stage where recycling must be minimal, the recorded colocalization should reflect anterograde transport rather than recycling.

The inability to record significant colocalization of UapA or SynA with the ERGIC/early-Golgi marker (SedV) in our movies (time frames of 6 s) might reflect very rapid (<6 s) passage of both cargoes from early secretory compartments. This assumption seemed the most logical, as at least SynA, which colocalizes to the late-Golgi, should 'necessarily' pass via the early-Golgi. Although we were unable to

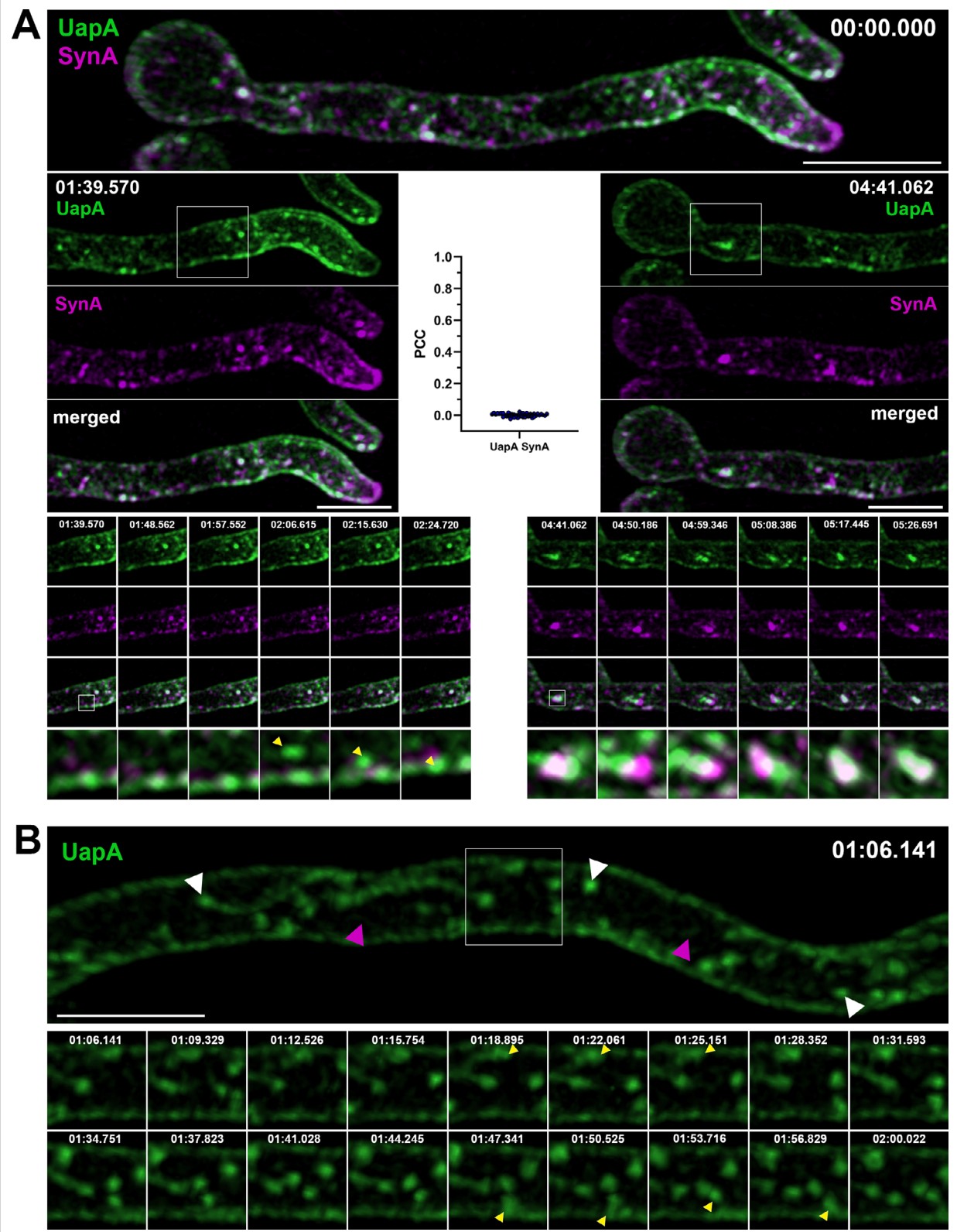

**Figure 2.** Synchronously co-expressed UapA and SynA mark distinct dynamic secretory compartments on their way to the plasma membrane (PM) (high-speed spinning-disc confocal microscopy). (**A**) Maximal intensity projections of deconvolved snap shots of UapA and SynA trafficking dynamics, extracted from a 7-min video (*Figure 2—video 1*), after synchronous derepression of transcription for 3 hr, using ultra-fast spinning-disc confocal microscopy. UapA-GFP has a non-polarized distribution to the PM, while mCherry-SynA is mostly localized in the apical area. UapA structures moving

*Figure 2 continued on next page*

*Figure 2 continued*

laterally toward the PM are indicated with yellow arrow heads (left bottom panel). Cytoplasmic UapA and SynA structures do not colocalize significantly (Pearson's correlation coefficient [PCC] = 0.01, $n$ = 52), apart from a small number of doubly labeled, mostly non-cortical puncta (0.33 ± 0.03 µm in diameter) with low-range mobility (right bottom panel). (**B**) Maximal intensity projections of deconvolved snap shots of UapA trafficking dynamics, extracted from a 3-min video (*Figure 2—video 2*), showing that UapA-GFP labels cytoplasmic oscillating thread structures decorated by pearl-like foci (0.32 ± 0.02 µm – white arrow heads) as well as a very faint vesicular/tubular network (magenta arrow heads). Pearl-like structures (yellow arrow heads) moving toward the PM are estimated to reach their destination in 12–18 s. Scale bars: 5 µm.

The online version of this article includes the following video(s) for figure 2:

**Figure 2—video 1.** 7-min video of de novo UapA-GFP versus mCherry-SynA after 3 hr of derepression.

https://elifesciences.org/articles/103355/figures#fig2video1

**Figure 2—video 2.** 3-min video of de novo UapA-GFP trafficking dynamics after 3 hr of derepression.

https://elifesciences.org/articles/103355/figures#fig2video2

**Figure 2—video 3.** 2-min video of de novo mCherry-SynA after 3 hr of derepression.

https://elifesciences.org/articles/103355/figures#fig2video3

estimate the relevant contribution of early-Golgi in the trafficking of distinct cargoes, a challenge that would require more ultra-fast high-resolution microscopy and/or synchronization of controlled ER-exit (see later), our findings, already provided strong evidence that the two cargoes studied traffic via distinct routes. SynA seems to be secreted via the conventional Golgi-dependent mechanism, while UapA is probably sorted to the PM from an early secretory compartment, ultimately bypassing the need for Golgi maturation. This assumption is strengthened by subsequent experiments.

## UapA and SynA sorting to the PM is differentially dependent on COPII components

The evidence that UapA and SynA follow distinguishable trafficking routes, which are differentially dependent on Golgi maturation, leads to the hypothesis of existence of functionally discrete ERES/COPII or ERGIC-type subpopulations that might be involved in the generation of distinct trafficking carriers (vesicles or tubules). One subpopulation (the 'canonical') drives the budding of carriers containing polarized cargoes like SynA or other growth-related proteins to the Golgi, whereas the second type of carrier (the 'unconventional') bypasses the Golgi to sort non-polarized cargoes like UapA or other transporters to the PM (*Dimou and Diallinas, 2020*; *Dimou et al., 2022*; *Dimou et al., 2020*). To further investigate the nature of distinct cargo carriers originating from the ER in *A. nidulans*, we examined the role of key proteins necessary for COPII formation and proper cargo ER-exit using our system of synchronous co-expression of SynA and UapA in the same cell. We specifically tested the role of the major COPII structural components SarA$^{Sar1}$, Sec24, Sec13, and Sec31, as well as Sec12, a guanine nucleotide exchange factor (GEF) that regulates SarA. Notice that SarA, Sec24, and Sec13 have been previously shown to be essential for growth, while Sec24 or Sec13 have also been found critical for proper trafficking of both polarized and non-polarized cargoes (*Hernández-González et al., 2015*; *Dimou et al., 2022*; *Dimou et al., 2020*). Noticeably, however, these studies have been performed in different strains for each cargo so that a direct comparison of the relative contribution of COPII components in the trafficking of distinct cargoes has not been possible.

Here, we analyzed the relative role of COPII components in the trafficking of UapA versus SynA using newly constructed thiamine-repressible alleles for SarA, Sec31 and Sec12 (for strain construction see Materials and methods), as well as previously made repressible alleles of Sec24 and Sec13 (*thiA$_p$-sec24* and *thiA$_p$-sec13*; *Dimou et al., 2020*). All repressible alleles were introduced by genetic crossing in the strain co-expressing UapA-GFP and mCherry-SynA via the regulatable *alcA* promoter. Western blot analysis confirmed that we could tightly repress the expression of SarA, Sec24, Sec13, and Sec31 (*Figure 4A*). Sec12 could not be detected in western blots, probably due to an apparent high turnover or modification of its terminal regions. As expected, thiamine repression of COPII-related genes led to lack (*sarA*, *sec24*, *sec13*, or *sec31*) or dramatically reduced (*sec12*) colony growth in the respective strains (*Figure 4B*). Using these strains, we followed the relative subcellular localization of UapA versus SynA (*Figure 4C*).

Repression of SarA, Sec24, Sec31, and Sec13 led to total retention of SynA at the ER network in ~100% of cells. UapA trafficking was also totally blocked upon SarA or Sec31 repression (100% of

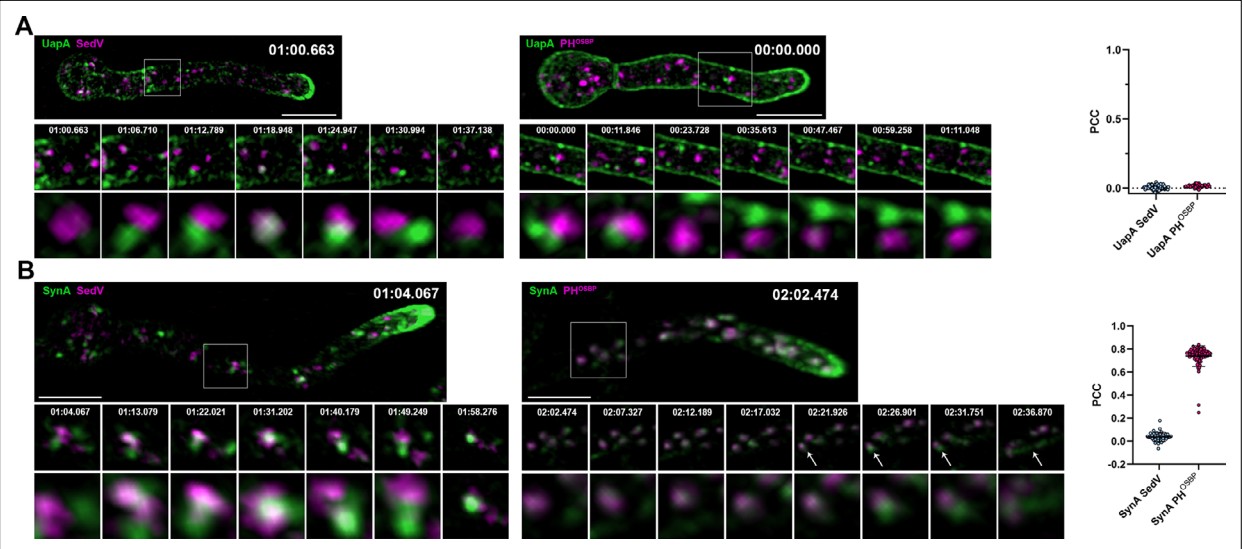

**Figure 3.** Unlike SynA, dynamically secreted UapA, does not colocalize with late-Golgi marker. (**A**) Maximal intensity projections of deconvolved snap shots of deconvolved videos (**Figure 3—videos 1 and 2**) in young hyphae after derepression of transcription of UapA-GFP for 3 hr. Colocalization study of UapA with the ERGIC/early-Golgi marker mCherry-SedV (left panel) and with the late-Golgi/trans-Golgi network (TGN) marker PH$^{OSBP}$-mRFP (right panel). Neosynthesized UapA-GFP does not colocalize significantly with any of the Golgi markers as indicated by the Pearson's correlation coefficient (PCC) (0.01, $n$ = 100 UapA-SedV; 0.01, $n$ = 35 UapA-PH$^{OSBP}$). A small number of colocalized structures of UapA-GFP with mCherry-SedV is observed lasting ~6 s, without being statistically significant. (**B**) Maximal intensity projections of deconvolved snap shots of videos (**Figure 3—videos 3 and 4**) in young hyphae after derepression of transcription of GFP-SynA for 3 hr. Colocalization study of SynA with the ERGIC/early-Golgi marker mCherry-SedV (left panel) and with the late-Golgi/TGN marker PH$^{OSBP}$-mRFP (right panel). Unlike UapA-GFP, neosynthesized GFP-SynA colocalizes significantly with the late-Golgi/TGN marker PH$^{OSBP}$ (PCC = 0.74, $n$ = 76, p < 0.0001) and very little with the ERGIC/early-Golgi marker SedV (PCC = 0.04, $n$ = 40). Notice that after exiting the TGN, fast moving SynA puncta are directed to the apical area of the hyphae (white arrow heads – right panel). Scale bars: 5 μm.

The online version of this article includes the following video and figure supplement(s) for figure 3:

**Figure supplement 1.** Super Resolution Radial Fluctuation (SRRF) imaging of colocalization of SynA with the late-Golgi marker PH$^{osbp}$ in fixed cells.

**Figure 3—video 1.** 10-min video of mCherry-SedV versus de novo UapA-GFP after 3 hr of derepression.
https://elifesciences.org/articles/103355/figures#fig3video1

**Figure 3—video 2.** 6-min video of mRFP-PH$^{OSBP}$ versus de novo UapA-GFP after 3 hr of derepression.
https://elifesciences.org/articles/103355/figures#fig3video2

**Figure 3—video 3.** 5-min video of mCherry-SedV versus de novo GFP-SynA after 3 hr of derepression.
https://elifesciences.org/articles/103355/figures#fig3video3

**Figure 3—video 4.** 6-min video of mRFP-PH$^{osbp}$ versus de novo GFP-SynA after 3 hr of derepression.
https://elifesciences.org/articles/103355/figures#fig3video4

cells) and was also highly blocked when Sec24 was repressed (~76% of cells). The partial independence of UapA localization on *sec24* repression might be due to the existence of a Sec24 homolog, namely LstA, in *A. nidulans* (**Dimou et al., 2022**). LstA is known to function as a cargo selection adaptor, similar to Sec24, for bulky or large oligomeric cargoes (**Dimou et al., 2022**). Notice that UapA is indeed a large cargo (i.e., the UapA dimer has 28 transmembrane segments), while SynA is a relatively small cargo (single transmembrane segment).

Interestingly, UapA trafficking to the PM was not efficiently blocked upon Sec13 repression in most samples (~80% of cells). This was quite surprising given that *sec13* is an absolutely essential gene for growth and SynA trafficking. Given that *A. nidulans* has no other Sec13-like proteins, this suggests that Sec13 is partially dispensable for UapA exit from the ER. The dispensability of Sec13 for UapA trafficking in most cells overrules our previous conclusion, which considered it as an essential protein for translocation of UapA or other transporters to the PM (**Dimou et al., 2020**; **Dimou et al., 2022**). Apparent discrepancy in our previous and present findings is probably due to the fact that previous studies have been performed in strains that separately expressed the two cargoes, so that a direct *relative* comparison could not have been performed. This shows the validity of our new system of

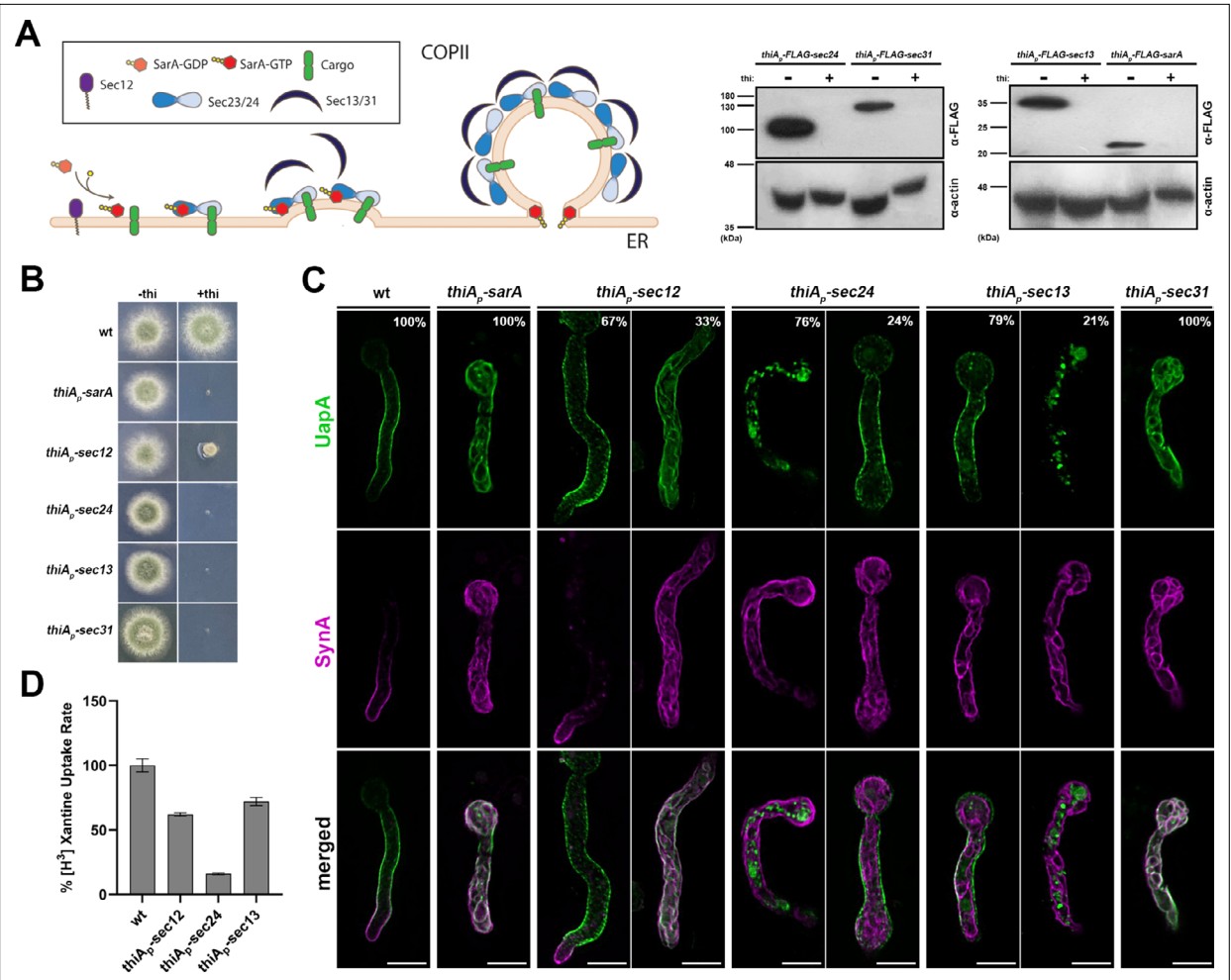

**Figure 4.** UapA and SynA sorting to the plasma membrane (PM) is differentially dependent on COPII components. (**A**) Overview of COPII formation at ER-Exit Sites (ERES) (left panel). Western blot analysis of COPII-repressible alleles *sarA*, *sec24*, *sec13*, and *sec31* (right panel). More specifically, COPII proteins were expressed under the tightly repressed *thiA_p* in the presence and absence of thiamine in the medium. Notice that in the absence of thiamine, all COPII proteins (SarA, Sec24, Sec13, and Sec31) are expressed and detected using anti-FLAG antibody. However, when thiamine is added in the medium, *thiA_p* is repressed and the respective proteins are not expressed to an extent not detectable in western blot analysis. Equal loading and protein steady-state levels are normalized against the amount of actin, detected with an anti-actin antibody. Sec12 detection was impossible even in the absence of thiamine. (**B**) Growth test analysis of strains expressing the repressible COPII components. In the absence of thiamine (−thi) all strains form colonies similar to wt, while in the presence of thiamine (+thi) strains do not grow at all (*thiA_p-sarA*, *thiA_p-sec24*, *thiA_p-sec13*, and *thiA_p-sec31*) or form dramatically reduced colonies (*thiA_p-sec12*). (**C**) Maximal intensity projections of deconvolved snap shots showing the subcellular localization of de novo UapA-GFP and mCherry-SynA using the *alcA_p-uapA-gfp/alcA_p-mCherry-synA* strain in different genetic backgrounds repressible for conventional COPII formation. In wt background UapA was sorted non-polarly to the PM, while SynA was localized in the apical area of the PM. Repression of SarA, Sec24, Sec13, and Sec31 led to total retention of SynA at the ER network in 100% of cells. UapA trafficking was also totally blocked upon SarA or Sec31 repression (100% of cells), and also efficiently blocked when Sec24 was repressed (~76% of cells). Notably, UapA trafficking was not blocked upon Sec13 repression in most cells (79%). Notice also that when the trafficking of SynA is blocked, SynA partitions in the ER membranes, whereas in the fraction of cells where UapA translocation to the PM is blocked, the protein forms aggregates (e.g., in 24% of cells repressed for *sec24*, and 21% of cells repressed for *sec13*). UapA aggregation might be to the fact that it is a large homodimeric protein with 28 transmembrane segments that oligomerizes further upon translocation into the ER membrane, apparently causing ER stress and turnover. Sec12 repression led to a moderate simultaneous negative effect on the sorting of UapA and SynA to the PM (33%) The number of cells used for quantitative analysis upon repressed conditions was n = 101 for SarA, n = 158 for Sec12, n = 163 for Sec24, n = 256 for Sec13, and n = 126 for Sec31, respectively. (**D**) Relative $^3$H-xanthine transport rates of UapA in genetic backgrounds repressible for COPII components Sec12, Sec24, and Sec13 expressed as percentages of initial uptake rates compared to the wt. UapA-mediated transport was very low when Sec24 was repressed (~15% of wt) but remained relatively high when Sec13 or Sec12 was repressed (~65–72% of wt). Results are averages of three measurements for each concentration point. Scale bars: 5 μm.

The online version of this article includes the following source data for figure 4:

**Source data 1.** Original files for western blot analysis displayed in *Figure 4A*.

**Source data 2.** PDF file containing original western blots for *Figure 4A*, indicating the relevant bands and treatments for each strain.

synchronous co-expression of two cargoes in the same cell, especially when the role of a trafficking factor on different cargoes is variable, as is the case of Sec13.

Given the differential requirement of Sec13 and other COPII components on UapA translocation to the PM, we further addressed the degree of relative contribution of Sec13 and Sec24 by measuring the capacity of transport of radiolabeled xanthine (i.e., substrate of UapA) in strains expressing the respective $thiA_p$-repressible alleles. In other words, we took advantage of the fact that UapA, our model cargo, is a native transporter whose activity is directly related to proper integration to the PM. *Figure 4D* shows that UapA-mediated xanthine transport was very low when Sec24 was repressed (~15% of wt) but remained relatively high when Sec13 was repressed (~72% of wt or non-repressed levels). This confirmed the high degree of dispensability of Sec13 for UapA functional translocation to the PM, relative to the other COPII components.

Finally, Sec12 repression of transcription was shown to lead to a moderate negative effect on the trafficking of both UapA and SynA to the PM (~33–38%, $n = 116$, $t = 4$), compatible with a less profound effect on growth (see *Figure 4B*). Sec12 is very probably present at extremely very low levels when expressed from its native promoter under the condition of our experiment (minimal media). This is supported by our recent proteomic analysis, performed under similar conditions, which failed to detect the Sec12 protein, unlike all other COPII components see *Dimou et al., 2021*, but also by cellular studies of the group of M.A. Peñalva, who failed to detect Sec12 tagged with GFP (*Bravo-Plaza et al., 2019*). Given that repression of *sec12* transcription via the *thiA* promoter still allows 68% of cells to secrete normally both SynA and UapA, while 32% of cells are blocked in the trafficking of both cargoes, suggests that in most cells either SarA can catalyze the exchange of GDP for GTP without Sec12, maybe through a cryptic GEF, or that very small amounts of Sec12 remaining after repression are sufficient for significant SarA activation. Additionally, in yeast, immune detection of Sec12 has been possible only in cells harboring *sec12* on a multicopy plasmid, suggesting its low abundance in wild-type cells (*Nakano et al., 1988*). Whichever scenario is true, Sec12, similar to SarA, is not critical for distinguishing Golgi-dependent from Golgi-independent routes, as both cargoes are affected similarly.

Overall, the above findings suggest that while SynA proper trafficking requires a 'canonical' COPII functional formation, UapA seems to be able to be secreted to the PM in the absence of Sec13, which is a major component of the outer coat of COPII. The differential role of Sec13 on specific cargo trafficking is discussed further later.

## Genetic block in COPII formation or repression of COPI activity traps UapA and SynA in distinct ERES

Live imaging of newly synthesized membrane cargoes in the ER is challenging because these proteins are continuously being synthesized and traffic rapidly from the ER to their final subcellular destination. Thus, trafficking cargoes might not be detected at the ER because their concentration is too low for detection by fluorescence microscopy. To overcome this technical limitation, the thermosensitive COPII allele *sec31-1* has been used in yeast as a genetic tool to reversibly retain the newly synthesized fluorescent-tagged cargoes in the ER or aberrant ERES in a temperature-dependent fashion (*Castillon et al., 2009*; *Rodriguez-Gallardo et al., 2021*). Through this system, synchronous cargo ER-exit can also be restored by shifting the cells to a permissive temperature (25°C). Here, we used a Sec31 thermosensitive approach for achieving a synchronized dual block of UapA and SynA and investigate whether the two cargoes are trapped in similar or distinct compartments. We constructed several *A. nidulans* Sec31 mutant strains (see Materials and methods) and identified one that showed a thermosensitive growth phenotype at 42°C (*Figure 5A*). The relevant mutation, named *sec31ts-AP*, was introduced by genetic crossing to the strain synchronously co-expressing UapA and SynA, and performed live microscopy.

As shown in *Figure 5B* (left panel), when spore germination takes place at 42°C, both cargoes fail to reach the PM, as expected given the essentiality of Sec31, shown also earlier in this work (see *Figure 4*). This confirms that the specific *sec31ts* allele used has severally inactivated Sec31 function at higher than the physiological temperatures, in line with growth tests shown in *Figure 5A*. Importantly, UapA and SynA marked very distinct areas of a membranous network typical of the ER (see merged panel). When cells were shifted from 42 to 25°C and observed after 0–45 min, the picture changed (*Figure 5B*, right panel). More specifically, both cargoes progressively marked puncta on or

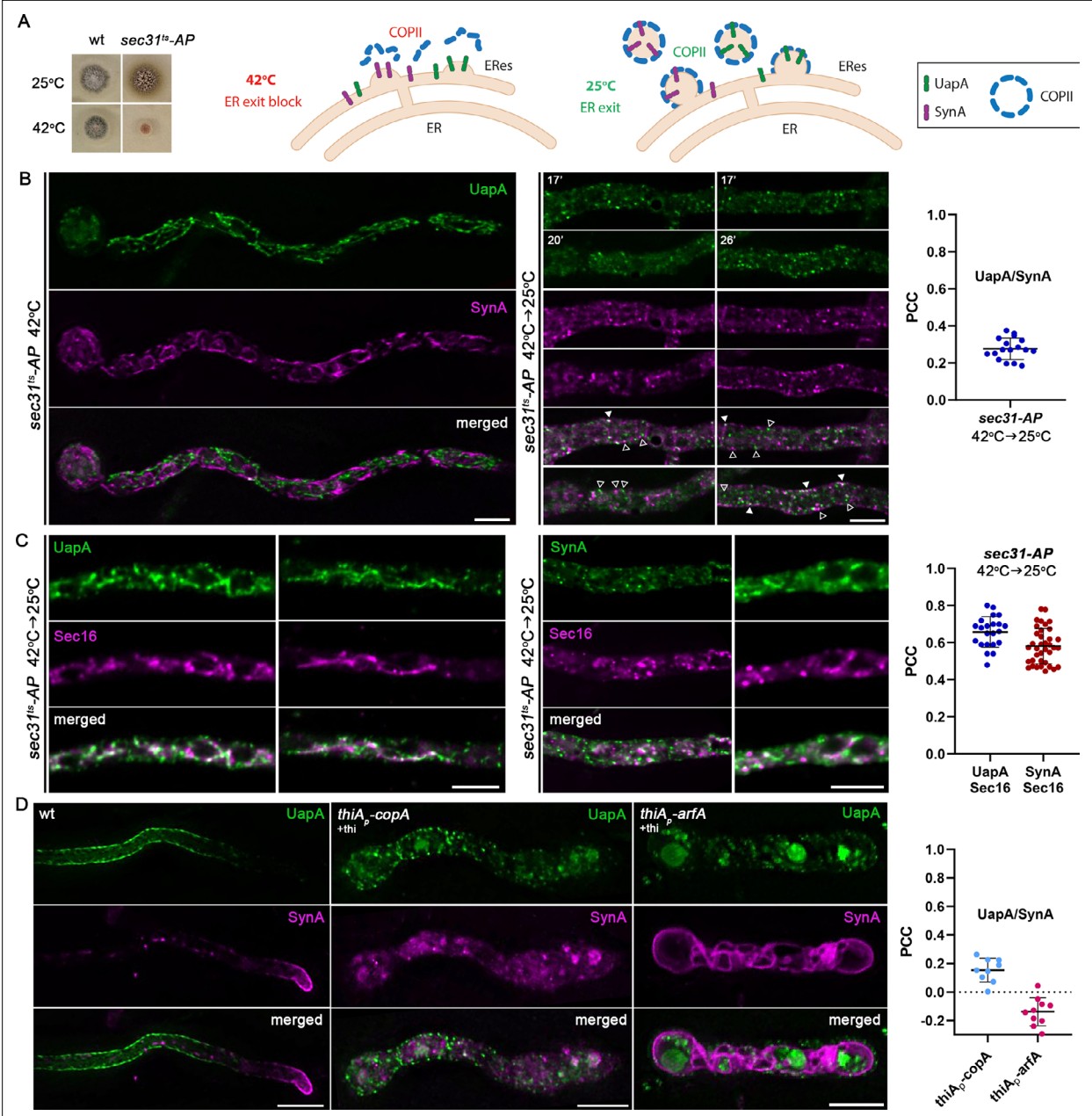

**Figure 5.** Genetic block in COPII formation or repression of COPI activity traps UapA and SynA in distinct ER-associated aggregates. (**A**) Growth test analysis of *sec31ts-AP* at 25 and 42°C, pH 6.8, compared to an isogenic wild-type (wt) control strain. Notice the severe growth defect of *sec31ts-AP* at 42°C (left panel). Strategy for achieving a synchronized accumulation/release of de novo made UapA and SynA at ER-Exit Sites (ERES) (right panel). (**B**) Maximal intensity projections of deconvolved z-stacks, using a *sec31ts-AP* strain co-expressing UapA-GFP and mCherry-SynA. At the restrictive temperature (42°C), both cargoes are retained in the ER marking distinct areas of a membranous network (merged image). Upon shifting down to the permissive temperature (42→25°C), ER export is restored and cargoes are progressively incorporated into ERES-like structures. UapA and SynA appear mostly in distinct puncta, showing a low degree of colocalization (open arrowheads, non-colocalizing dots; white arrowheads, colocalizing dots), as illustrated also by the calculation of Pearson's correlation coefficient (PCC) on the right (PCC = 0.28 ± 0.06, n = 17). (**C**) Maximal intensity projections of deconvolved z-stacks, using *sec31ts-AP* strains co-expressing UapA-GFP/Sec16-mCherry or GFP-SynA/Sec16-mCherry upon a shift from 42 to 25°C. Both UapA and SynA exhibited significant colocalization with the ERES marker Sec16 (PCC = 0.66 ± 0.08, n = 22 UapA-Sec16; PCC = 0.58 ± 0.09, n = 37 SynA-Sec16) strongly suggesting that UapA and SynA are recruited to spatially distinct ERES. (**D**) Maximal intensity projections of deconvolved z-stacks, using strains co-expressing UapA-GFP and mCherry-SynA under conditions where *copA* or *arfA* transcription is repressed by addition of thiamine (+thi). De novo synthesis of cargoes takes place after full repression of CopA or ArfA is achieved (>16 hr). Repression of either CopA or ArfA abolishes the translocation of both cargoes to the plasma membrane (PM). Notice that upon CopA repression, UapA and SynA collapse in distinct internal structures, similar to those when the *sec31ts-AP* strain was shifted from 42 to 25°C. Quantification of colocalization (right panel) when *copA* or *arfA* transcription is

*Figure 5 continued on next page*

*Figure 5 continued*

repressed, by calculating PCC, shows clear non-colocalization of UapA with SynA (PCC = 0.15 ± 0.08, *n* = 9 in *thiA_p_-copA* and PCC = −0.14 ± 0.1, *n* = 10 in *thiA_p_-arfA*). Scale bars: 5 µm.

closely associated with the ER membranous network, often associated with rings reflecting nuclear ER, but also in cortical regions next to the PM. Most notably, UapA and SynA puncta did not colocalize significantly (PCC = 0.28 ± 0.06, *n* = 17). The distribution, size, and rather static nature of UapA and SynA puncta strongly suggested that these reflect distinct ERES, as they appear when marked with FP-tagged COPII components (e.g., Sec24; *Dimou et al., 2020*). Thus, when Sec31 is inactivated (42°C) and proper ERES cannot be formed, UapA and SynA collapse in extensive parts of the ER network, but when Sec31 is progressively re-activated by a shift at 25°C, stable COPII complexes start being formed, giving rise to apparently proper ERES and normal trafficking. In this series of events, UapA and SynA never colocalized significantly, which in turn suggested that the two cargoes, after post-translational translocation to the ER, partition and exit from distinct ERES, subsequently forming spatially distinguishable secretory compartments.

To further define the identity of the ER-associated puncta marked with UapA or SynA, we constructed *sec31^ts^* strains that co-express either UapA-GFP or GFP-SynA with a standard marker of ERES, namely Sec16, tagged with mCherry (for strain construction see Material and methods). *Figure 5C* shows the localization of UapA-GFP/Sec16-mCherry or GFP-SynA/Sec16-mCherry in the presence of *sec31^ts^* mutation after growth in a non-permissive temperature (42°C) and transfer to permissive temperature (25°C). Both cargoes show significant colocalization with Sec16 (PCC ≈ 0.6) in clearly defined puncta, as those detected in *Figure 5B*. These findings strongly suggested that upon block-and-release of ER-exit, both cargoes are sorted to ERES. Ideally, for showing directly the existence of distinct cargo-specific ERES, a three-color approach, using red, green, and blue fluorescent tags, would be needed. However, the BFP-tagged Sec16 failed to give a specific and strong signal for performing colocalization studies with UapA-GFP and mCherry-SynA (not shown). Nevertheless, given that UapA and SynA mark distinct compartments, as clearly shown in *Figure 5B*, our results strongly suggest that the two cargoes are recruited in spatially distinct ERES.

To further follow the parallel fate of UapA and SynA carriers exiting the ER, we also tried to detect their dynamic localization when immediate downstream steps in their secretion were blocked. More specifically, we tested their localization when COPI function is conditionally downregulated by repression of transcription of either CopA^Cop1^ or ArfA^Arf1/2^. CopA is an essential core component of the COPI complex, while ArfA is an essential ADP-ribosylation GTPase that regulates anterograde coated vesicle formation by recruiting COPI at the ERES–early-Golgi interface (*Weigel et al., 2021*; *Shomron et al., 2021*). We have previously constructed *thiA_p_*-repressible alleles of CopA and ArfA and shown that these affect trafficking of UapA or SynA, albeit in experiments performed in separate strains (*Georgiou et al., 2023*). Here, we introduced *thiA_p_-copA* and *thiA_p_-arfA* alleles in the dual strain synchronously co-expressing UapA and SynA. As shown in *Figure 5D*, repression of either CopA or ArfA, abolished translocation of these cargoes to the PM. Whether this signifies a direct block in anterograde transport or alternatively CopA/ArfA are needed to recycle proteins from ERES-derived carriers (or ERGIC) back to the ER, and thus indirectly affecting anterograde transport, is a still a debatable issue. Noticeably, the picture obtained upon CopA repression was rather similar to what has been observed when cargo trafficking was followed in the background of the *sec31^ts^* strain shifted from 42 to 25°C. More specifically, we observed distinct, little colocalizing (PCC = 0.15 ± 0.08, *n* = 9), UapA or SynA cytosolic puncta associated with the ER network. On the other hand, upon ArfA repression, the effect on cargo trafficking seemed more dramatic compared to CopA repression or *sec31^ts^*, with UapA sorted into vacuoles and SynA collapsed all over the ER membrane. This might be related to the fact the ArfA has a pleiotropic effect on other steps in cargo trafficking, besides recruitment of COPI. Notably, the ER membranes where SynA collapses upon ArfA repression is distinct from the structures marked with UapA (PCC = −0.14 ± 0.1, *n* = 10).

Overall, the findings described above show that UapA and SynA are trapped in distinct ERES and possibly downstream secretory compartments upon functional blocks in COPII or COPI formation.

## Golgi maturation and conventional post-Golgi vesicular secretion are redundant for UapA localization to the PM

Using a similar approach as that for studying the role of COPII components, we followed the synchronous co-expression and trafficking of de novo made UapA and SynA in mutant backgrounds where we could repress the transcription of genes encoding essential Golgi proteins or conventional post-Golgi secretion (*Pantazopoulou, 2016*; *Pinar and Peñalva, 2017*; *Pantazopoulou and Peñalva, 2009*; *Pantazopoulou and Glick, 2019*; *Nakano, 2022*). The role of Golgi proteins studied were (1) the ERGIC/early-Golgi proteins SedV$^{Sed5}$ (Q-SNARE) and GeaA$^{Gea1}$ (GEF for Arf GTPases), (2) the late-Golgi/TGN protein HypB$^{Sec7}$ (GEF for of Arf GTPases), (3) the RabO$^{Ypt1/Rab1}$ (GTPase involved in both early- and late-Golgi functioning), (4) RabE$^{Ypt31/32/Rab11}$ (GTPase essential for generation of post-Golgi vesicles from the TGN and cargo recycling from the PM), and (5) and AP-1$^{σ/Aps1}$ subunit of the clathrin adaptor complex essential for the generation and fission of TGN-derived vesicular carriers, but also for retrograde traffic in the late secretory pathway (*Casler et al., 2022*; *Robinson et al., 2024*). *thiA$_p$*-repressible alleles of the aforementioned Golgi proteins (*Dimou et al., 2020*) were introduced by genetic crossing in the strain synchronously co-expressing UapA-GFP and mCherry-SynA. As expected, the resulting *A. nidulans* strains could not form colonies when Golgi proteins were repressed, apart from the *thiA$_p$-hypB* mutant which produced small compact colonies (*Figure 6*).

After having established overnight repression of transcription of the Golgi proteins, we imaged the subcellular steady-state localization of de novo made SynA versus UapA (*Figure 6C*), Our results showed that in all cells repressed for SedV, GeaA, RabO, RabE, or AP-1$^σ$, and ~60% of cells repressed for HypB, SynA totally failed to accumulate in the PM of the apical region, and instead labeled cytoplasmic foci or large aggregates. These findings were in full agreement with previous reports supporting that SynA is a membrane cargo that traffics exclusively via the conventional Golgi-dependent vesicular secretion mechanism. In contrast to SynA, UapA translocated to the PM when several key Golgi proteins were repressed. More specifically, in GeaA, HypB, RabO, or AP-1$^σ$ repressible strains UapA translocation to the PM was practically unaffected in all cells (~100%). The non-essentiality of HypB for UapA trafficking, but not for SynA, was also in line with the colocalization of SynA, but not of UapA, with the late-Golgi/TGN marker, as shown earlier in *Figure 3* and *Figure 3—videos 1–4*. On the other hand, two proteins among those tested, namely SedV and RabE, had a partial but prominent negative effect on UapA translocation to the PM. Repression of SedV led to a block in UapA trafficking to the PM in ~65% of cells, while in ~35% of cells UapA was normally localized to the PM. The partial dependence of UapA trafficking on SedV seems to be in line with the low relative colocalization detected in *Figure 3*. Upon RabE repression, a significant fraction of UapA could translocate to the PM in ~75% of cells, but in this case, the same cells also showed numerous large cytoplasmic structures. These structures are probably UapA membrane aggregates, as they showed very little colocalization with Golgi markers (PCC = 0.17 ± 0.07, $n$ = 6 with SedV and PCC = 0.26 ± 0.08, $n$ = 11 with PH$^{OSBP}$; see *Figure 6—figure supplement 1*).

To validate further and quantify the effect of selected Golgi proteins on UapA localization to the PM, and particularly evaluate the partial role of SedV and RabE, we performed radiolabeled xanthine uptake assays. As shown in *Figure 6D*, we detected high levels of UapA activity (i.e., transport of radiolabeled xanthine) in strains repressed for HypB and AP-1$^σ$ (83–100% of the transport rate of wt), whereas transport activity was reduced to 27% and 19% of the wt when RabE and SedV were repressed, respectively. This confirmed a partial dependence of UapA trafficking on RabE and SedV. A possible explanation for the indirect role of RabE and SedV on UapA trafficking is discussed later.

To further address the importance of Golgi proteins in the trafficking of SynA and UapA we also used the standard approach of pharmacological inactivation of Golgi by addition of Brefeldin A (BFA). Notice that BFA does not affect early secretion steps, as ERES or ERGIC formation seems intact (*Lippincott-Schwartz et al., 1990*; *Saraste and Svensson, 1991*; *Füllekrug et al., 1997*). Our results showed that addition of BFA for 60 min after derepression of cargoes for 90 min, blocks the trafficking of SynA in all cells (~100%), leading to the trapping of this cargo into large Golgi aggregates (known as 'brefeldin bodies'), but has only a very modest effect on UapA translocation to the PM (*Figure 6E*). Notice that the images of SynA versus UapA localization in the presence of BFA are very similar with those observed when the early-Golgi protein GeaA, which is a target of BFA, was repressed (see white head arrows in *Figure 6C*).

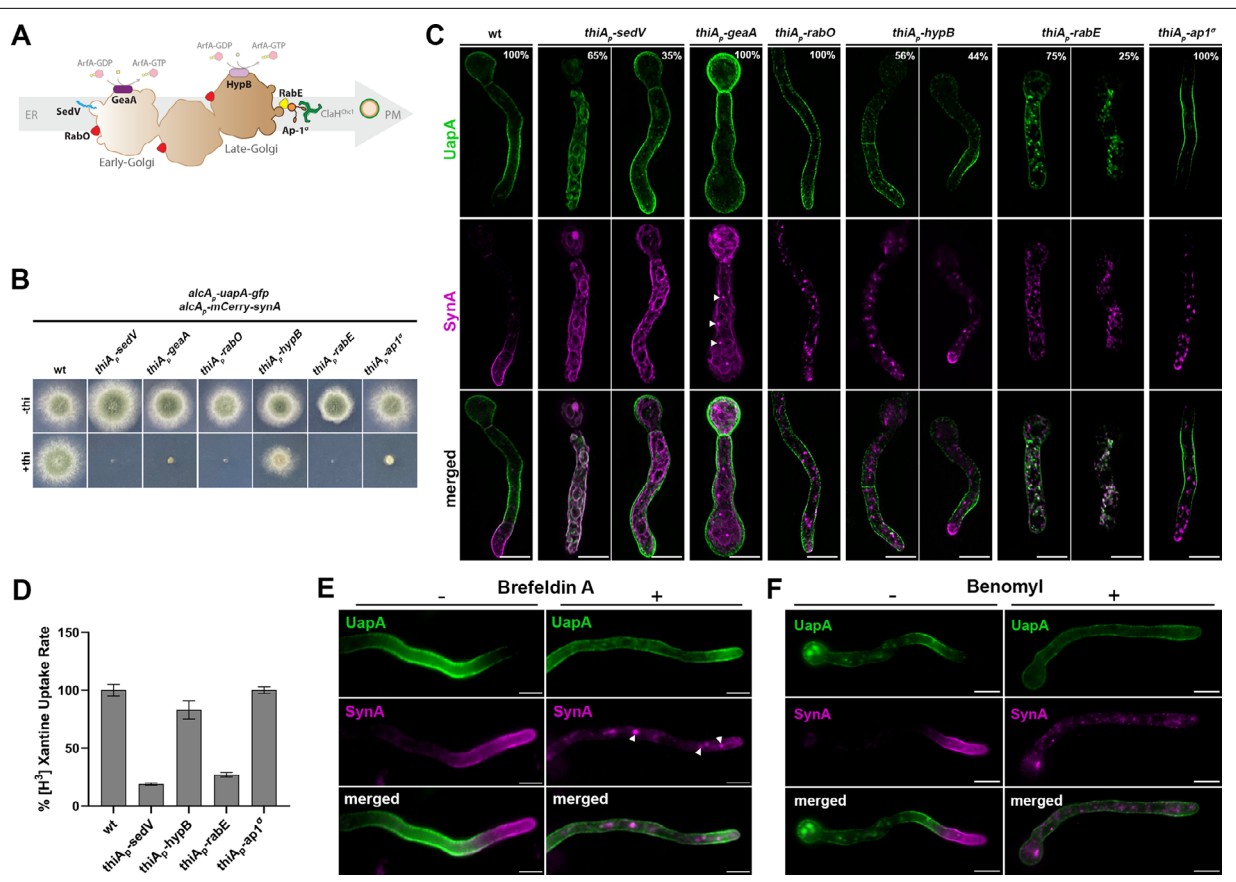

**Figure 6.** Golgi maturation and conventional post-Golgi vesicular secretion are not needed for UapA localization to the plasma membrane (PM). (**A**) Schematic depiction of Golgi maturation and post-Golgi secretion. (**B**) Growth test analysis of strains expressing early/late/post-Golgi repressible alleles *sedV*, *geaA*, *rabO*, *hypB*, *rabE*, or *ap1ᵟ* . In the absence of thiamine (−thi) all strains form colonies similar to wt, while in the presence of thiamine (+thi) strains could not form colonies when Golgi proteins were repressed, apart from the *thiAₚ-hypB* mutant which produced a small compact colony. (**C**) Maximal intensity projections of deconvolved snap shots showing the subcellular localization of de novo UapA-GFP and mCherry-SynA using the *alcAₚ-uapA-gfp/alcAₚ-mCherry-synA* strain in different genetic backgrounds repressible for conventional Golgi-dependent secretion. In wt background SynA was localized to the apical area of the PM, while UapA was sorted non-polarly to the subapical PM. Repression of SedV, GeaA, RabO, RabE, and Ap1ᵟ led to total retention of SynA in cytoplasmic structures, while repression of HypB abolished SynA trafficking in 56% of cells. Notice that upon repression of SedV and GeaA, SynA mostly labeled the ER network, indicating the close dynamic association of the ER with the ERGIC/early-Golgi. In contrast to SynA, UapA reached the PM when several essential ERGIC/early-Golgi and post-Golgi were repressed. More specifically, in GeaA, RabO, HypB, and Ap1ᵟ repressible strains UapA translocated to the PM in 100% of cells. Repression of SedV and RabE had a partial negative effect on UapA trafficking. In particular, lack of SedV led to block of UapA traffic in 65% of cells, while upon RabE repression, a significant fraction of UapA could translocate to the PM in ~75% of cells, but in this case, the same cells also showed a significant number of large cytoplasmic structures. The number of cells used for quantitative analysis upon repressed conditions was n = 266 for SedV, n = 201 for GeaA, n = 104 for RabO, n = 159 for HypB, n = 219 for RabE, and n = 190 for Ap1ᵟ, respectively. (**D**) Relative $^3$H-xanthine transport rates of UapA in genetic backgrounds repressible for SedV, HypB, RabE, and Ap1ᵟ expressed as percentages of initial uptake rates compared to the wt. UapA-mediated transport was very high when HypB or Ap1ᵟ were repressed (83–100% of wt), while transport activity was reduced to 27% and 19% of the wt when RabE and SedV were repressed, respectively. Results are averages of three measurements for each concentration point. (**E**) Subcellular localization of de novo UapA-GFP and mCherry-SynA in wt background upon treatment with the Golgi inhibitor Brefeldin A (BFA) after 90 min of derepression of cargoes. In the absence of BFA, UapA and SynA were localized properly in the PM. Addition of BFA for 60 min after derepression blocked SynA trafficking leading to its trapping into large Golgi aggregates (brefeldin bodies), while UapA could still reach the PM (upper panel). Notice that cytoplasmic SynA structures upon BFA addition are similar with those obtained when the early-Golgi protein GeaA, which is a target of BFA, was repressed (white arrow heads, lower panel). (**F**) Subcellular localization of de novo UapA-GFP and mCherry-SynA in wt background upon treatment with the anti-microtubule drug Benomyl for 90 min after derepression of cargoes. Notice that microtubule depolymerization is dispensable for UapA trafficking, while it is absolutely necessary for the proper sorting of SynA to the apical PM. Addition of Benomyl blocks SynA in cytoplasmic structures, while UapA translocation to the PM is unaffected. Scale bars: 5 μm.

The online version of this article includes the following figure supplement(s) for figure 6:

**Figure supplement 1.** UapA membrane aggregates upon RabE repression do not colocalize with early- or late-Golgi markers.

Finally, we tested the effect of microtubule depolymerization on SynA versus UapA traffic by adding benomyl in the sample of cells after 90 min of cargo derepression and before microscopic imaging. Our results showed that proper trafficking of SynA to the apical PM was totally abolished, as expected for a conventional cargo, while UapA translocates normally to the PM, in all cells, similar to the control without benomyl (*Figure 6F*).

Overall, our findings confirmed that in cells exhibiting a total block in SynA conventional secretion due to repression of Golgi maturation, UapA could still translocate to the PM.

## UapA translocation to the PM occurs in the absence of SNAREs essential for conventional fusion of ER-derived carriers to the cis-Golgi

Specific fusion of ER-derived vesicles with target Golgi membranes requires the pairing of SNARE proteins embedded in vesicles (i.e., v-SNAREs) to SNAREs anchored in the target membrane (i.e., t-SNAREs). This pairing forms *trans*-SNARE complexes that bridge lipid bilayers (*Weber et al., 1998*; *Südhof and Rothman, 2009*). In *S. cerevisiae*, the SNARE proteins Sed5, Bos1, Bet1, and Sec22 have been shown to form a basic *trans*-SNARE complex required for vesicular traffic from the ER to the Golgi apparatus (*Shim et al., 1991*; *Dascher et al., 1991*; *Hardwick and Pelham, 1992*; *Parlati et al., 2000*). Importantly, two additional yeast SNAREs, Ykt6 and Sft1, have also been shown to play essential roles in cargo trafficking via their interaction with Sed5 at the ER–Golgi interface (*Kweon et al., 2003*; *Adnan et al., 2019*). Given the difficulty in assigning rigorously whether a SNARE is anchored in vesicles or target membranes, an alternative classification is based mostly on their sequence similarity but also the presence of either Arg (R) or Gln (Q) residue in the motif that is needed for SNARE self-assemblage into a stable four-helix bundle in the *trans*-complex (*Bock et al., 2001*). By this classification, Sec22 and Ykt6 are R-SNAREs, while Sed5, Bos1, Bet1 or Sft1 are Qa-, Qb-, and Qc-SNAREs, respectively. Commonly, a v-SNARE is an R-SNARE and t-SNAREs are Q-SNAREs. Notice however that Bet1, Bos1, and Sft1 despite lacking an R residue are also annotated as v-SNARES (https://www.yeastgenome.org/), and that Bet1 and Sft1, although phylogenetically classified as Q-SNAREs, they possess polar residues (T/S or D) other than the critical Q at their SNARE motifs. In most cases, the *trans*-SNARE complex is formed by a single R-SNARE and three structurally and phylogenetically distinct Qa/b/c-SNAREs. In yeast, null mutants of the Q-SNAREs Sed5, Bos1, Bet1 or Sft1 and of the R-SNARE Ykt6 are inviable, whereas the deletion of the R-SNARE Sec22 is viable but affects the trafficking of specific cargoes. The viability of Δ*sec22* is thought to be due to Ykt6, which seems to be a multitask R-SNARE involved in vesicular fusion occurring at several trafficking compartments, as for example, at the ER–Golgi interface, within Golgi, during endocytosis, or in vacuolar sorting (*Kweon et al., 2003*). Orthologs of the yeast R- and Q-SNAREs (e.g., synaptobrevin, syntaxins, or SNAP-25) are localized and act similarly in mammalian and plant cells.

In the previous section, we showed a partial dependence of UapA trafficking on SedV. To investigate further how SedV might affect UapA trafficking, we tested the role of putative partners of SedV in the *trans*-SNARE complex that mediates fusion of ER/ERGIC-derived vesicles to early-Golgi cisternae. Based on the known SNARE interactions regulating ER to Golgi sorting in *S. cerevisiae* outlined above, SedV (Qa-SNARE) might interact with orthologs of Bos1 (Qb-SNARE), Bet1 or Sft1 (Qc-SNAREs), and Sec22 or Ykt6 (R-SNAREs). Thus, we identified the relative genes and constructed repressible (*thiA_p-bos1*, *thiA_p-bet1*, *thiA_p-ykt6*, and *thiA_p-sft1*) or total deletion (Δ*sec22*) alleles of all aforementioned SNAREs. Strains carrying *thiA_p-sft1* and *thiA_p-ykt6* alleles could not form colonies in the presence of thiamine but grew normally in the absence of thiamine. The Δ*sec22* mutant and *thiA_p-bos1* and *thiA_p-bet1* strains also showed a significant growth defect under repressed conditions, but less prominent compared to that where Ykt6 or Sft1 were repressed (*Figure 7A*). The significant negative effect on growth of strains expressing *thiA_p*-driven transcription of the SNARE genes strongly suggested that repression of the respective SNARE proteins was successful.

The repressible or null alleles of *bet1*, *bos1*, *sft1*, *ykt6*, and *sec22* were introduced by genetic crossing to the strain co-expressing UapA and SynA. *Figure 7B* shows the steady-state subcellular localization of neosynthesized UapA and SynA upon specific SNARE knockdown or knockout. SynA proper trafficking was severely mitigated in ~100% of cells upon repression of Sft1 or Ykt6, while it was little affected when Bet1 was repressed (reduced accumulation at the apex), and not affected when Bos1 was repressed or in the Δ*sec22* mutant. Notice also that when Sft1 or Ykt6 are repressed hyphae become somewhat wider compared to wt or strains downregulated for Bos1, Bet1, or Sec22. Most

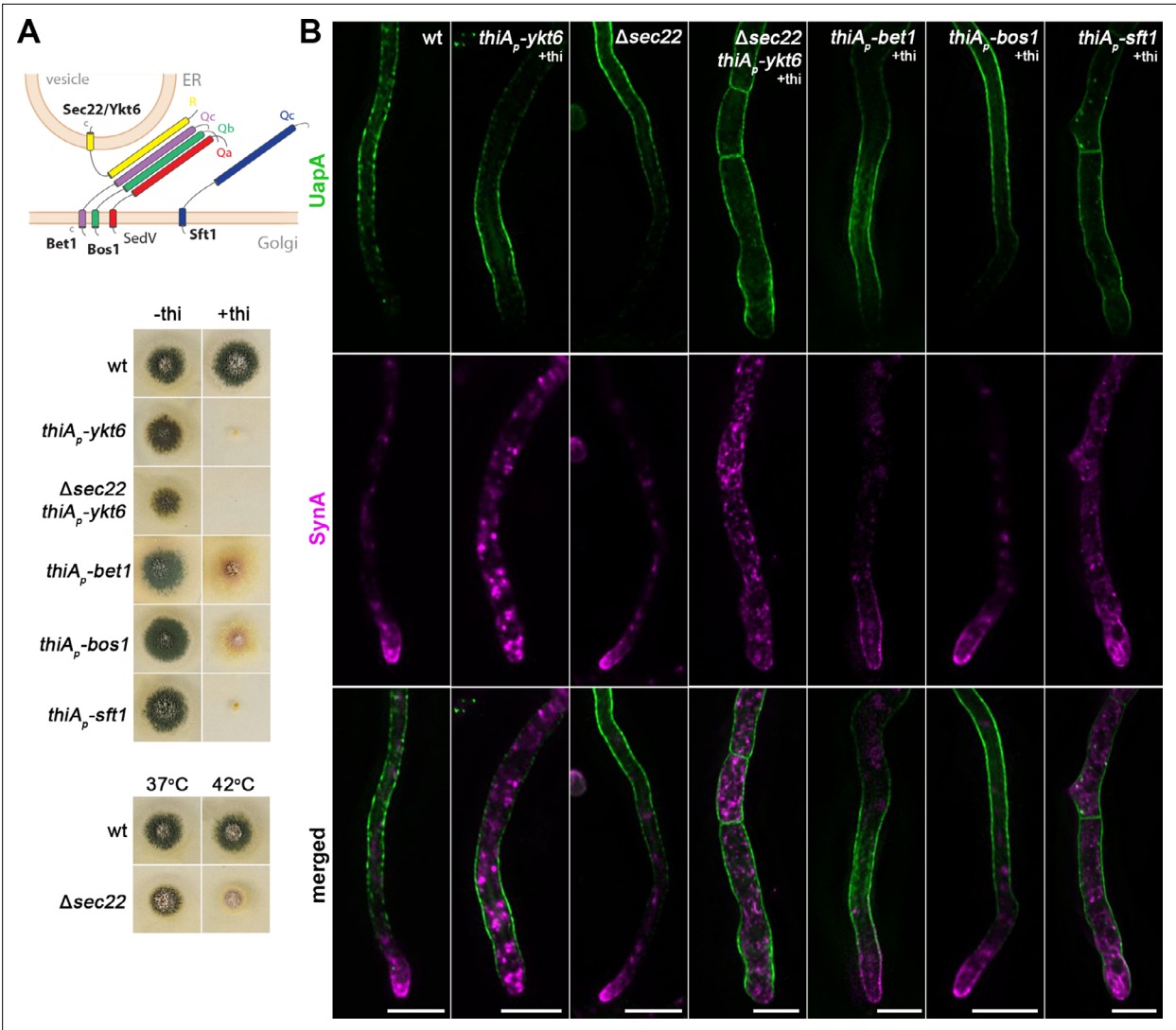

**Figure 7.** UapA translocation to the plasma membrane (PM) occurs independently of SNAREs Ykt6, Sec22, Bet1, and Bos1. (**A**) Schematic illustration of the SNARE complex mediating membrane fusion of ER-derived vesicles to Golgi cisternae (upper panel). Growth phenotypes of null or repressible mutants of SNAREs involved in ER-to-Golgi trafficking (lower panels). In the absence of thiamine from the growth medium (derepressed conditions), the corresponding strains grow nearly as an isogenic wild-type control strain. In the presence of thiamine to the growth medium, *thiA_p-ykt6*, *Δsec22 thiA_p-ykt6*, and *thiA_p-Sft11* do not form colonies, while *thiAp-bet1* and *thiAp-bos1* form a slow-growing colony with reduced conidiospore production. Growth test analysis of *Δsec22* at 37 and 42°C, pH 6.8, compared to an isogenic wild-type (wt) control strain, showing a temperature-dependent growth defect at 42°C. (**B**) Maximal intensity projections of deconvolved z-stacks, using ER-to-Golgi SNARE mutant strains co-expressing UapA-GFP and mCherry-SynA. UapA translocation to the PM is not affected in any of the SNARE mutants tested, even in the double mutant strain lacking both Ykt6 and Sec22. SynA trafficking to the hyphal apex is not impaired upon Bos1 or Bet1 repression or Sec22 deletion, but is abolished upon Ykt6 or Sft1 repression, and in the double mutant *thiA_p-ykt6 Δsec22*. Scale bars: 5 μm.

importantly, UapA trafficking to the PM was not affected in any of the five SNARE mutants tested. Given that Ykt6 and Sec22 might have redundant activities, being R-SNARES, in respect to cargo trafficking, we also constructed the double mutant strain *Δsec22 thiA_p-ykt6*, 'lacking' both these SNAREs and followed UapA versus SynA trafficking (see *Figure 7A, B*). As expected, SynA proper trafficking was abolished in *Δsec22 thiA_p-ykt6*, but still UapA translocation to the PM remained unaffected.

In conclusion, our findings showed that unlike SynA trafficking, which required Ykt6 and Sft1, we did not identify any possible SNARE partners of SedV, among those conventionally operating at the ER to Golgi vesicular transport, as important for UapA trafficking. This indicated that UapA trafficking might involve unprecedented non-canonical interactions of SNAREs, other than the ones normally involved in the secretion of polarized cargoes, as further supported in the next section.

## UapA translocation to the PM requires the SsoA-Sec9 (Qa/b-SNARE) complex, but not its conventional partner SynA (R-SNARE) or the exocyst effector RabD

The final step in the biogenesis of transmembrane proteins destined to the cell membrane consists of tethering and fusion of vesicular carriers, originating from the TGN, to the PM. In yeast, this occurs via the action of the hetero-octameric exocyst complex (a tethering complex) and the *trans*-interaction of Qa/b-SNAREs Sso1/2 and Sec9 with vesicular R-SNAREs Snc1/Snc2 (*Couve and Gerst, 1994*; *Katz and Brennwald, 2000*). In this complex, the Snc1 and Snc2 (synaptobrevin/VAMP family) seem to function as the v-SNAREs of Golgi/TGN-derived secretory vesicles. Vesicular fusion to the PM also requires RabD$^{Sec4}$, an essential Rab GTPase, that recruits secretory vesicles to the exocyst tethering complex (*Guo et al., 1999*; *Pantazopoulou et al., 2014*).

To study the final steps of translocation of UapA to the PM versus that of polarized cargoes, we constructed mutant strains where the *A. nidulans* genes encoding the *trans*-SNARE complex proteins

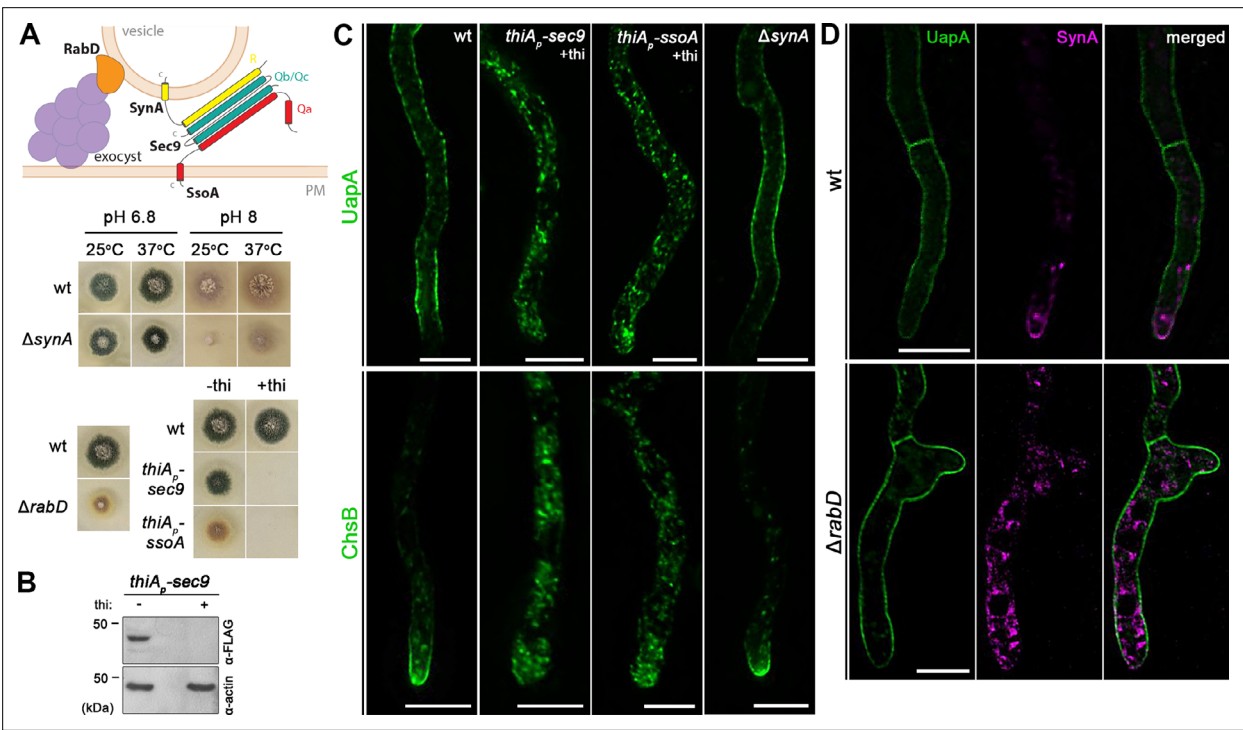

**Figure 8.** UapA translocation to the plasma membrane (PM) requires the SsoA-Sec9 Q-SNARE complex, but not the R-SNARE SynA or the exocyst effector RabD. (**A**) Schematic illustration of the SNARE complex mediating membrane fusion of secretory vesicles to the PM (upper panel). Growth phenotypes of null or repressible mutants of SNAREs involved in vesicular fusion to the PM (lower panels). Growth test analysis of ΔsynA at 25 and 37°C, at pH 6.8 and 8.0, compared to an isogenic wild-type (wt) control strain. Notice the reduced colony growth of ΔsynA at 25°C, pH 8.0. Growth test analysis of ΔrabD at 37°C, pH 6.8 compared to an isogenic wild-type (wt) control strain, showing a markedly reduced colony size resulting from rabD knock-out. Growth test analysis of the *thiA$_p$-sec9* and *thiA$_p$-ssoA* strains compared to a wild-type control strain in the presence (+thi) or absence (−thi) of thiamine from the media, showing that Sec9 or SsoA repression impedes colony formation. (**B**) Western blot analysis of Sec9 using the anti-FLAG antibody. In the absence of thiamine (−thi) from the growth medium, Sec9 is expressed, while upon addition of thiamine (+thi) at the onset of conidiospore germination (*ab initio* repression), the expression of these proteins is tightly repressed. Equal loading and protein steady-state levels are normalized against the amount of actin, detected with an anti-actin antibody. (**C**) Maximal intensity projections of deconvolved z-stacks, using strains expressing UapA-GFP or GFP-ChsB in genetic backgrounds where exocytic SNAREs are repressed or deleted. Repression of Sec9 or SsoA abolishes the translocation of UapA and ChsB to the PM, while SynA deletion has no effect on the trafficking of both cargoes to the PM. (**D**) Maximal intensity projections of deconvolved z-stacks, using a ΔrabD strain co-expressing UapA-GFP and mCherry-SynA. Deletion of RabD impairs the translocation of SynA to the apical PM, which remains trapped in cytosolic membrane aggregates, while UapA trafficking to the PM is not affected. Scale bars: 5 μm.

The online version of this article includes the following source data and figure supplement(s) for figure 8:

**Source data 1.** Original files for western blot analysis displayed in *Figure 8B*.

**Source data 2.** PDF file containing original western blots for *Figure 8B*, indicating the relevant bands and treatments for each strain.

**Figure supplement 1.** Triple R-SNARE knockout Δsec22 ΔnyvA ΔsynA does not affect UapA or ChsB trafficking to the plasma membrane (PM).

SynA$^{Snc1/2}$, SsoA$^{Sso1/2}$, and Sec9, or the exocyst effector RabD$^{Sec4}$, were deleted or could be repressed. Surprisingly, unlike the double deleted Δ*snc1*Δ*snc2* yeast mutant, which is inviable, the Δ*synA* mutant showed only a minor growth defect at alkaline pH at 25°C (*Figure 8A*). On the other hand, strains harboring *thiA$_p$-sec9* or *thiA$_p$-ssoA* alleles could not form colonies on a Petri dish supplemented with thiamine (*Figure 8A*). Western blot analysis confirmed that the expression of Sec9 could be tightly repressed in the presence of thiamine (*Figure 8B*). SsoA could not be detected via terminal FP tagging in western blots, but efficient repression is strongly supported by the inability of the relevant strain to form colonies in the presence of thiamine. Finally, the deletion of *rabD* (Δ*rabD*) had a prominent morphological defect coupled with altered colony morphology, as also previously described by others (*Pantazopoulou et al., 2014*; *Figure 8A*). The Δ*synA*, *thiA$_p$-sec9*, and *thiA$_p$-ssoA* alleles were introduced, by genetic crossing, into strains expressing either UapA-GFP or GFP-ChsB, while the Δ*rabD* allele was introduced by genetic crossing to the strain co-expressing UapA and SynA. ChsB is a chitin synthase that here was used as a standard polarly localized apical cargo, similar to SynA (*Martzoukou et al., 2018*; *Dimou et al., 2020*).

Live cell imagining showed that repression of SsoA or Sec9 (*Figure 8C*) blocked the translocation of both ChsB and UapA to the PM, as these cargoes marked cytoplasmic puncta. In contrast, both cargoes reached the PM normally in the Δ*synA* mutant. Given the dispensability of SynA for growth and cargo translocation to the PM, we also tested whether NyvA$^{Nyv1}$, a fungal specific R-SNARE (v-SNARE), might replace the function of SynA. NyvA homologs are thought to be involved in cargo sorting to vacuoles, vacuolar homotypic fusion, and autophagy, but have also been detected in the PM (*Nichols et al., 1997*; *Gupta and Brent Heath, 2002*; *Kuratsu et al., 2007*). However, the Δ*nyvA* null mutation did not affect growth and has not affected trafficking of UapA or ChsB. To further test a possible additive or compensatory role of different v-SNARES, we also constructed and tested the triple combination of Δ*nyvA* Δ*synA* Δ*sec22*. The combination of these three R-SNAREs did not affect neither UapA nor ChsB trafficking to the PM (*Figure 8—figure supplement 1*).

Lastly, in the strain deleted for RabD (Δ*rabD*) UapA localization to the PM was not affected, while SynA translocation to the apical PM (or that of ChsB; not shown) was significantly blocked (*Figure 8D*).

## Discussion

Contrasting the general concept of conventional Golgi-dependent trafficking of membrane cargoes, we recently provided evidence that in *A. nidulans* several non-polarized PM proteins, mostly transporters, translocate to the PM via a Golgi maturation-independent trafficking route. Evidence for Golgi-independence was obtained by colocalization studies coupled with a genetic system where proteins essential for Golgi function or post-Golgi secretion could be knocked-down by transcriptional repression (*Dimou and Diallinas, 2020*; *Dimou et al., 2020*; *Dimou et al., 2022*). Despite its apparent advantages, our previous system for evaluating the contribution of Golgi and post-Golgi secretion in cargo biogenesis suffered from the fact that trafficking was evaluated in separate strains for each specific cargo, and in some cases the cargoes were expressed through different promoters. This endangered quantitative misevaluation of results obtained due to non-homogenous efficiency of transcriptional repression of Golgi-related genes in all cells, a problem most obvious in cases where the trafficking factor repressed had a partial role on cargo secretion (e.g., SedV or RabE).

Here, we improved our genetic system to be able to follow in parallel the trafficking of specific neosynthesized polarized (SynA) and non-polarized (UapA) cargoes, *synchronously co-expressed* via the *same* regulatable promoter in the *same* cell, and in addition, we managed to synchronize cargo accumulation and exit from the ER by using a novel *sec31$^{ts}$* allele. These approaches allowed us to compare and rigorously quantify the *relative* dynamic localization and the effect of secretory proteins on the trafficking of neosynthesized UapA versus SynA, irrespective of developmental stages and only in cells where repression of genes essential for conventional secretion route was totally arrested (i.e., cells that show a total block in SynA localization to the apical membrane). We thus obtained strong evidence that UapA and SynA, representing examples of *native* polarized versus non-polarized PM cargoes, follow distinct trafficking routes in *A. nidulans*.

The evidence concerning the independence of UapA trafficking from essential proteins necessary for Golgi maturation and post-Golgi vesicular secretion was very clear specifically in genetic backgrounds repressible for GeaA (early/medial Golgi), HypB (late-Golgi), RabO (early- and late-Golgi), or AP-1$^\sigma$ (post-Golgi). In these strains proper UapA translocation to the PM was observed in nearly 100%

cells, which at the same time showed a total block in SynA trafficking. Furthermore, proper UapA trafficking was shown not to be affected by the Golgi-disrupting drug BFA or the tubulin depolymerizing agent benomyl, unlike the absolute negative effect of these drugs on SynA trafficking. These findings strengthen our previous results pointing to a mechanism that bypasses Golgi maturation and post-Golgi vesicular secretion for transporters and other non-polarized proteins. Importantly, the present work goes further in presenting novel findings on the specific contribution of COPII, COPI, specific SNAREs and exocyst activity in the trafficking of UapA relative to polarized cargoes (SynA or ChsB).

Highlights of our new findings include (1) the minor contribution of Sec13, an essential COPII component for growth, to the trafficking of UapA, (2) the partitioning of UapA and SynA to distinct ERES and downstream early secretory compartments, similar to intermediate compartments (ICs) or ERGIC upon reversible Sec31 inactivation or repression of CopA activity, and (3) the non-involvement in proper UapA trafficking of essential ER-to-Golgi or Golgi-to-PM SNAREs other than Sec9 and SsoA. Altogether, these results strongly support the existence of a specific trafficking route for UapA, independent of conventional Golgi maturation, post-Golgi secretion, or canonical SNARE interactions. The partial but prominent negative effect of the knockdown of SedV or RabE on Golgi-independent UapA trafficking is very interesting and needs to be studied further, as it points to new roles, direct or indirect, of these proteins, in addition to their involvement in conventional secretion.

The discovery of spatially and dynamically distinct secretory compartments for UapA and SynA, coupled with the independence of UapA proper trafficking from late-Golgi or post-Golgi secretory mechanisms, or canonical SNARE interactions and exocyst recruitment, strongly suggest the existence of a distinct cargo-specific type of vesicular or tubular carrier that drives UapA, and very possibly other non-polarized cargoes, to the PM from specific ERES. Distinct ERES generating specific COPII subpopulations have been shown to be formed in vitro and in vivo for specific glycosyl-phosphatidylinositol (GPI)-anchored proteins or other cargoes, such as hexose or amino acid transporters (*Muñiz et al., 2001*; *Watanabe et al., 2008*; *Castillon et al., 2009*; *Castillon et al., 2011*; *Rodriguez-Gallardo et al., 2021*), but also interleukin-1b (*Zhang et al., 2015*; *Zhang et al., 2020*). The early distinction of trafficking routes in these cases seems to be driven via specific cargo associations with chaperones/adaptors (e.g., p24 or Erv14 proteins), specific ER-exit motifs on cargoes, and/or differential dependence on COPII accessory proteins (Sec12 or Sec16), which all seem to lead to partitioning into distinct ERES prior to budding. Most interestingly, a Rab1-dependent, Golgi-independent, pathway connecting directly a pre-Golgi IC with the PM has been reported in polarized neuroendocrine PC12 cells (*Sannerud et al., 2006*; *Saraste and Marie, 2018*). This IC, which is spatially segregated from canonical ERGIC membranes, seems to act as a cargo-specific tubular network that delivers, via Golgi-bypass, PM components in developing neurites. Given its well established essentiality in conventional secretion, this work further reveals a new role of Rab1 GTPase in regulating Golgi-bypass via interactions with apparently non-canonical effectors. In line with these findings, it was also shown that Golgi and canonical Rab1 effectors, but not the ERES generating unconventional ICs, are excluded from neurites. Notably, in our case, RabO, which is the fungal homolog of RaB1, was found to be dispensable for Golgi-bypass. Despite this important difference, we cannot avoid noticing a possible analogy of the role of Golgi-bypass to the development and maintenance of highly polarized cells, such as filamentous fungi and neuronal cells.

Several more cargoes undergoing Golgi-independent trafficking in mammalian cells might also be secreted via specific ERES or ICs exiting the ER (*Rabouille, 2017*; *Gee et al., 2018*; *Camus et al., 2020*; *Zhang et al., 2020*; *Kemal et al., 2022*; *Sun et al., 2024*). In fact, distinct early secretory ICs have been reported in plants, where two types of ERGIC-type subpopulations have been identified (*Fougère et al., 2023*). One subpopulation forms a dynamic reticulated tubulo-vesicular network, generated independently from Golgi maturation, while the other subpopulation is more stable, more cisterna-like, and dependent on Golgi. Apparently, the generation of two types of ERGIC serves the trafficking of structurally and functionally different cargoes. However, the distinct ER-exit routes detected in plants have not been related to Golgi-sorting versus Golgi-bypass.

Possibly related to the existence of distinct ERES, UapA trafficking, unlike that of SynA, can take place in the absence of Sec13. A rationalization of this finding might come from reports showing that intrinsic cargo properties and/or the mode of cargo oligomerization might lead to surface-crowding effects and influence the bending of membranes, which in turn might specify differential needs for different COPII components, and particularly for Sec13, as reflected in the isolation of several mutants

bypassing the need for Sec13 in yeast (*Elrod-Erickson and Kaiser, 1996*; *Fatal et al., 2002*; *Copic et al., 2012*; *Springer et al., 2014*). Most notably, Sec13 has been shown, through both genetic and biochemical approaches, to serve the rigidification of the COPII cage, which in turn leads to an increase in its membrane-bending capacity, especially needed when specific cargoes tend to elicit the opposing curvature toward the lumen of the ER (*Copic et al., 2012*). Interestingly, a requirement for Sec13 can also be bypassed when non-essential membrane cargo proteins are depleted (e.g., GPIs or p24-family proteins). These proteins cause a concave negative curvature of the membrane into the ER lumen, which to be inverted necessitates Sec13 (*D'Arcangelo et al., 2015*). Thus, for specific cargoes, Sec31 is mechanistically sufficient to generate COPII vesicles in the absence of Sec13, albeit with reduced efficiency, as we also recorded here for UapA. UapA is known to form a tight dimer, necessary for its transport activity, and genetic evidence suggests that UapA dimers might further oligomerize to achieve concentrative exit from the ER (*Martzoukou et al., 2015*; *Kourkoulou et al., 2019*). It is thus probable that UapA forms large oligomers that might elicit sufficient positive curvature to rigidify the COPII cage and thus promote ER-exit, even in the absence of Sec13. One can further speculate that the presence or absence of Sec13 in specific ERES, driven by properties of the cargo itself, might affect the size, composition, and membrane scaffolding of the emerging cargo carriers, and thus dictate their downstream trafficking routes.

Related to the aforementioned ideas, it was interesting to find out that none of standard R-SNAREs (v-SNAREs) known to pack in canonical COPII vesicles destined to the early-Golgi was found to be essential for proper UapA trafficking. The absence of a canonical ER-Golgi R/Q-SNARE complex might be sufficient to diverge these carriers away from Golgi and instead promote fusion to proximal PM via a short-range 'hug-and-kiss' mechanism (*Kurokawa et al., 2014*). The physiological consequence and rationale of a mechanism bypassing late-Golgi functioning might be to uncouple the trafficking of cargoes localized all over the PM, serving nutrition and homeostasis, from microtubule-directed secretion of apical cargoes, which are essential for growth. Furthermore, in line with Golgi-bypass, fungal nutrient transporters, unlike those of mammals, are not glycosylated (*Zielinska et al., 2012*), so Golgi-bypass might serve to avoid fortuitous glycosylation of transporters and other cargoes that function as non-glycosylated proteins. Noticeably also, the trafficking of Chs1, a chitin synthase localized in growing hyphal tips but also in nascent septa and subapical spherical structures in the filamentous fungus *Neurospora crassa* has been found to be unaffected upon Benomyl and BFA treatment, which suggests a Golgi-bypass mechanism (*Sánchez-León et al., 2011*). As also discussed earlier, Golgi-bypass has been reported for several other mammalian or plant transmembrane cargoes (*Rabouille, 2017*; *Gee et al., 2018*; *Camus et al., 2020*; *Zhang et al., 2020*; *Dimou and Diallinas, 2020*; *Kemal et al., 2022*; *Sun et al., 2024*). However, most of these studies concern trafficking routes operating only under specific conditions or stress, and do not seem to reflect a major trafficking mechanism for the biogenesis of essential cargoes, as transporters or receptors, proposed through our work. Notice also that in *A. nidulans* transporters and apical cargoes use different endocytic mechanisms for their internalization, turnover or recycling (*Martzoukou et al., 2017*).

Despite the possibility that SedV might affect UapA trafficking indirectly, as discussed earlier, we also addressed the possibility of a direct role by investigating the role of all principal SNAREs in yeast known to functionally interact with Sed5 (Qa-SNARE), possibly forming the *trans*-SNARE complex necessary for transfer of cargoes from ERES/ERGIC to early-Golgi cisternae. We showed that none of these (Sec22, Ykt6, Bet1, Bos1, or Sft1) is critical for UapA translocation to the PM, while repression of SedV, Sft1, and Ykt6 totally blocks ER-exit and sorting of SynA to the apical membrane. We subsequently obtained evidence that although localization of UapA and apical cargoes (e.g., ChsB) to the PM requires the SsoA/Sec9 (Q-SNARE) complex, it is however independent of SynA, which is considered the canonical R-SNARE syntaxin partner of SsoA/Sec9 in both yeast and mammals. Since the knockdown of SynA had little effect on growth and there are no SynA paralogs in *A. nidulans*, this suggests that for most cargoes a 'non-canonical' v-SNARE (R-SNARE) might interact with SsoA-Sec9 at the PM in this fungus.

Overall, the only SNAREs found to be strictly necessary for UapA localization to the PM were SsoA and Sec9. This foresees the existence of a novel, unprecedented *trans*-interaction of these Qa/b/c-SNAREs with other SNARE partners not studied herein. Interestingly, there are reports showing that the 3Q:1R ratio might not be necessary the formation of a *trans*-SNARE complex. In these studies, genetic and biochemical assays have shown that SNARE complexes composed of four glutamines (4Q)

are fully functional for assembly in vitro and exocytic function in vivo. Thus, SNARE complexes may, in principle, be composed entirely of Q-SNAREs in non-canonical, context-dependent, combinations, for the trafficking of specific cargoes or under specific conditions (*Katz and Brennwald, 2000*; *Ossig et al., 2000*; *Parlati et al., 2002*). In line with this, some SNAREs, like Sed5, are known to possess remarkable plasticity in mediating distinct *trans*-SNARE complexes (*D'Souza et al., 2023*). In *A. nidulans,* as in other systems, there are additional Q-SNAREs related to ER to Golgi sorting of cargoes and essential for growth, not studied here (*Gupta and Brent Heath, 2002*; *Kuratsu et al., 2007*). These are homologs of the *S. cerevisiae* SNAREs: Ufe1 (Qa), Tlg2 (Qa), Sec20 (Qb), Gos1 (Qb), Vti1 (Qb), Use1 (Qc), and Tlg1 (Qc). Notice also that Sed5 has been shown to physically interact with several of these SNAREs (e.g., Gos1, Vti1, Use1 and Tlg1; https://www.yeastgenome.org/). Most interestingly, in a recent article it has been shown that hyperactive Sly1, a member of the Sec1/mammalian Unc-18 (SM) family of SNARE chaperones, can directly tether close-range vesicles with high curvature to the Qa-SNARE on the target organelle (*Duan et al., 2024*). This leads to the alternative and rather provocative possibility that cargo carriers bypassing the late-Golgi/TGN might fuse to the PM without any v-SNARE in them.

Notice also that in the filamentous fungus *Trichoderma reesei* growing apical and non-growing subapical hyphal compartments have been shown to possess spatially distinct ternary SNARE complexes at their PM, involving two SsoA-like paralogs (SSOI and SSOII) interacting with their SynA-like partner (SNCI). This finding indicated that there is more than one pathway for exocytosis in filamentous fungi, employing different surface SNARE proteins, at apical and subapical regions of fungal cells (*Valkonen et al., 2007*). Distinct exocytic routes in apical and subapical regions have also been suggested in older studies in filamentous fungi (*Harsay and Bretscher, 1995*; *Nykanen et al., 1997*; *Davis et al., 2000*). Future studies will be necessary to define the role of all SNAREs in respect to unconventional cargo trafficking in *A. nidulans*.

Lastly, we obtained indirect evidence, via knockout of RabD, suggesting that the exocyst is not needed for UapA localization in the PM. On the other hand, RabD was found to be essential for proper growth and SynA translocation to the apical membrane. Previous findings on the tethering role of the exocyst complex in *A. nidulans* showed that, similar to UapA, secretion of the non-polarized GPI-anchored protein EglC to the cell periphery and septa was little impaired in the absence of RabD (*Peñalva et al., 2020*). Notice also that while fungal exocyst proteins and associated effectors (e.g., Sec4 or Sec1) are polarly localized, the functionally relevant transmembrane SNARE SsoA is distributed all along the PM, despite a relatively higher abundance in the apical membrane (*Martzoukou et al., 2018*; *Dimou et al., 2020*). This suggests that the PM *trans*-SNARE complex might serve exocytic vesicular fusion at the subapical membrane of hyphae via an exocyst-independent non-canonical mechanism, as also suggested before (*Grote and Novick, 1999*).

We believe that our present work paves the way to the next obvious step in dissecting the mechanistic details of unconventional trafficking routes of transporters or other non-polarized cargoes in eukaryotes. This should necessitate the employment of more powerful high-speed live cell imaging at high spatial resolution, coupled with in vitro vesicle/tubule budding and fusion reconstitution assays. The coupling of such approaches to the unique genetic tools and mode of extreme polarized growth of *A. nidulans*, or other model filamentous fungi, is expected to enrich the study of membrane tracking in eukaryotes.

## Materials and methods
### Media, strains, growth conditions, and transformation
Standard complete and minimal media for *A. nidulans* were used (FGSC, http://www.fgsc.net). Media and chemical reagents were obtained from Sigma-Aldrich (Life Science Chemilab SA, Hellas) or Appli-Chem (Bioline Scientific SA, Hellas). Glucose 1% (wt/vol) or fructose 0.1% (wt/vol) was used as carbon source. $NH_4^+$ (di-ammonium tartrate) or $NaNO_3^-$ were used as nitrogen sources at 10 mM. Thiamine hydrochloride was used at a final concentration of 10–20 μM as a repressor of the $thiA_p$ promoter (*Dimou et al., 2020*) in microscopy or western blot analysis. *A. nidulans* transformation was performed by generating protoplasts from germinating conidiospores using TNO2A7 (*Nayak et al., 2006*) or other *nkuA* DNA helicase deficient strains, that allow in-locus integrations of gene fusions via auxotrophic complementation. Integrations of gene fusions with fluorescent tags (GFP/mRFP/mCherry),

promoter replacement fusions (*alcA$_p$*/*thiA$_p$*) or deletion cassettes were selected using the *A. nidulans* marker para-aminobenzoic acid synthase (*pabaA*, AN6615) or the *A. fumigatus* markers orotidine-5-phosphate-decarboxylase (AF*pyrG*, Afu2g0836), GTP-cyclohydrolase II (AF*riboB*, Afu1g13300) or a pyridoxine biosynthesis gene (AF*pyroA*, Afu5g08090), resulting in complementation of the relevant auxotrophies. Transformants were verified by PCR and growth test analysis. Combinations of mutations and fluorescent epitope-tagged strains were generated by standard genetic crossing and progeny analysis. *E. coli* strains used were DH5a. *A. nidulans* strains used are listed in *Supplementary file 2*.

## Nucleic acid manipulations and plasmid constructions

Genomic DNA extraction was performed as described in FGSC (http://www.fgsc.net). Plasmid preparation and DNA gel extraction were performed using the Nucleospin Plasmid and the Nucleospin Extract II kits (Macherey-Nagel, Lab Supplies Scientific SA, Hellas). Restriction enzymes were from Takara Bio (Lab Supplies Scientific SA, Hellas). DNA sequences were determined by Eurofins-Genomics (Vienna, Austria). Mutations were constructed by site-directed mutagenesis according to the instructions accompanying the Quik Change Site-Directed Mutagenesis Kit (Agilent Technologies, Stratagene). Conventional PCRs were performed with KAPA Taq DNA polymerase (Kapa Biosystems, Lab Supplies Scientific). High-fidelity amplification of products and site-directed mutagenesis were performed with Kapa HiFi polymerase (Kapa Biosystems, Lab Supplies Scientific). Gene cassettes were generated by sequential cloning of the relevant fragments in the pGEM-T plasmid (Promega), which served as template to PCR-amplify the relevant linear cassettes. The same vector was used for the construction of the *sec31* mutations, by changing alanine at position 1249 to valine, proline, serine or glycine, based on sequence alignment to the *S. cerevisiae sec31-1* mutation (*Salama et al., 1997*). Oligonucleotides used for cloning and site-directed mutagenesis purposes are listed in *Supplementary file 3*.

## Conditions used to repress–derepress cargo expression

For following the subcellular trafficking and localization of de novo made UapA-GFP and mCherry-SynA we used the regulatable *alcA* promoter combined with a repression-derepression setup analogous to the one described in *Dimou et al., 2020*. In brief, cargo expression was repressed by overnight growth (for 12–14 hr, at 25°C) in the presence of glucose as sole carbon source, and derepressed by a change to fructose for the following 1–8 hr of growth. For following the trafficking of the polarized cargo GFP-ChsB, we used its native promoter. In experiments aiming at repressing key trafficking proteins expressed from the *thiA* promoter, 10 μM thiamine was used throughout growth. In the case of *thiA$_p$-copA* and *thiA$_p$-arfA*, thiamine was not added *ab initio* in the culture, but after an 8 hr time-window without thiamine at the start of spore incubation, under conditions where cargo (UapA, SynA) expression was repressed, as described previously (*Georgiou et al., 2023*).

## Protein extraction and western blots

Total protein extraction was performed as previously described in *Dimou et al., 2020*, using dry mycelia from cultures grown in minimal media supplemented with NaNO$_3^-$ at 25°C. Total proteins (50 μg, estimated by Bradford assays) were separated in a 10% (wt/vol) polyacrylamide gel and were transferred on polyvinylidene fluoride (PVDF) membranes (GE Healthcare Life Sciences Amersham). Immunodetection was performed with an anti-FLAG antibody (Agrisera, AAS15 3037), or an anti-actin monoclonal (C4) antibody (SKU0869100-CF, MP Biomedicals Europe) and an HRP-linked antibody (7076, Cell Signalling Technology Inc). Blots were developed using the Lumi Sensor Chemiluminescent HRP Substrate kit (Genscript USA) and SuperRX Fuji medical X-Ray films (Fuji FILM Europe).

## Fluorescence microscopy

Conidiospores were incubated overnight in glass bottom 35 mm μ-dishes (ibidi, Lab Supplies Scientific SA, Hellas) in liquid minimal media, for 16–22 hr at 25°C, under conditions of transcriptional repression of cargoes expressed from the *alcA* promoter [1% (wt/vol) glucose] and repression of selected trafficking proteins expressed under the *thiA* promoter (10 μM thiamine). *Sec31$^{ts}$-AP* thermosensitive cells were incubated overnight at 42°C in glucose containing media. Transcriptional derepression of cargoes was followed through a shift in media containing fructose as a carbon source. Derepression

periods ranged from to 60 min to 4 hr, according to experiments. To test the effect of the thermo-sensitive *sec31^ts^-AP* on cargo ER accumulation/release, after a 2-hr derepression period at the restrictive temperature (42°C), cells were shifted down to the permissive temperature (25°C), to reverse the secretory block and allow cargoes to enter the ERES. Benomyl (Sigma-Aldrich) and BFA (Sigma-Aldrich) were used at 2.5 and 100 μg/ml final concentrations, respectively. Images from widefield microscopy were obtained using an inverted Zeiss Axio Observer Z1 equipped with the white light pE-400 Illumination System (https://www.coolled.com/products/pe-400) and a Hamamatsu ORCA-Flash 4 camera. All widefield z-stack images were deconvolved with Huygens Essential version 23.10 (Scientific Volume Imaging, The Netherlands, http://svi.nl). Contrast adjustment, area selection, color combining, and scale bar addition were made using the Fiji software (*Schindelin et al., 2012*). Images were further processed and annotated in Adobe Photoshop CS4 Extended version 11.0.2. Images and movies from ultra-fast spinning-disc microscopy were obtained using a motorized inverted Nikon Eclipse Ti2-E microscope equipped with the high-contrast multi-dimensional confocal-spinning disc system DragonFly 200 (Andor technology, https://andor.oxinst.com/products/dragonfly-confocal-microscope-system) and the Andor Sona sCMOS 4.2B-6 camera. Two of the four available lasers, 488 nm (cyan) for GFP excitation, 561 nm (yellow-green) for RFP/mCherry excitation, were used in combination with the emission filters 525/30 (510–540 nm) for GFP and 598/35 (580–615 nm) for RFP/mCherry, respectively. All images and videos were obtained using z-stacks, then deconvolved with the Fusion software version 2.3.0.44 (Andor – Oxford Instruments) and further processed in Imaris 10.1 (Andor – Oxford Instruments) for contrast adjustment, area selection, color combining and scale bar addition. SRRF imaging was performed in fixed cells using the confocal-spinning disc system DragonFly 200 and the SRRF-Stream^+ algorithm embedded in the Fusion software 2.3.0.44 (Andor – Oxford Instruments). SRRF-Stream^+ is based on the SRRF algorithm (*Gustafsson et al., 2016*) and delivers super-resolved images with a final resolution between 50 and 150 nm. In this experiment, images were obtained using 100 snap shots per channel (GFP and mRFP) per z-stack (57 z-stacks in total). Cells used for SRRF imaging were incubated overnight in glass bottom 35 mm μ-dishes in liquid minimal media, for 15 hr at 25°C, under conditions of transcriptional repression of cargo expressed from the *alcA* promoter [1% (wt/vol) glucose]. Transcriptional derepression of cargo was followed through a shift in medium containing fructose 0.1% (wt/vol) as a carbon source for 2 hr. Then, cells were firstly washed with phosphate-buffered saline (PBS), fixed in room temperature (RT) for 20 min using 4% formaldehyde (final concentration) and after washes with PBS, mounting medium (ibidi, Lab Supplies Scientific SA, Hellas) was added in fixed cells for their observation in the microscope. All images were further processed in Adobe Photoshop CS4 Extended version 11.0.2. Colocalization was calculated with the PCC using the BIOP JACoP plugin of Fiji. PCC was determined using maximal intensity projections of deconvolved z-stacks on manually traced regions of interest covering the hyphal cells. Data visualization and statistical analysis of PCCs were performed using the GraphPad Prism software. Confidence interval for one-sample *t*-test was set to 95%.

## Transport assays

Kinetic analysis of UapA into mutant backgrounds *thiA_p*-repressible for secretion-related proteins were measured by estimating uptake rates of [$^3$H]-xanthine uptake (40 Ci mmol$^{-1}$, Moravek Biochemicals, CA, USA), basically as previously described in *Krypotou and Diallinas, 2014*. In brief, [$^3$H]-xanthine uptake was assayed in *A. nidulans* conidiospores germinating for 16 hr at 25°C, at 140 rpm, in liquid MM supplemented with 10 mM thiamine and 1% (wt/vol) glucose for cargo repression, at pH 5.5. Then cells were shifted to MM supplemented with 10 mM thiamine and 0.1% (wt/vol) fructose, at pH 5.5, for 6 hr at 25°C, 140 rpm to derepress the cargo expression. Steady-state transport rates were measured on $10^7$ conidiospores/100 μl by incubation with concentration of 2.0 μM of [$^3$H]-xanthine at 37°C. All transport assays were carried out in two independent experiments and the measurements in triplicate. Standard deviation was <20%. Results were analyzed in GraphPad Prism software.

## Acknowledgements

We thank Sotiris Amillis for the construction of several strains. This work was supported by HFRI research grants 3112/KE18458 and 19067 to G Diallinas and a Fondation Santé fellowship to Sofia Dimou and Xenia Georgiou.

## Additional information

### Funding

| Funder | Grant reference number | Author |
|---|---|---|
| Hellenic Foundation for Research and Innovation | 3112/KE18458 | George Diallinas |
| Hellenic Foundation for Research and Innovation | 19067 | George Diallinas |
| Fondation Sante | scholarship | Sofia Dimou Xenia Georgiou |

The funders had no role in study design, data collection, and interpretation, or the decision to submit the work for publication.

### Author contributions

Georgia Maria Sagia, Data curation, Software, Formal analysis, Validation, Investigation, Visualization, Methodology, Writing – review and editing; Xenia Georgiou, Formal analysis, Investigation, Visualization; Georgios Chamilos, Software, Formal analysis; George Diallinas, Conceptualization, Data curation, Formal analysis, Supervision, Funding acquisition, Validation, Writing - original draft, Project administration, Writing – review and editing; Sofia Dimou, Data curation, Software, Formal analysis, Validation, Investigation, Visualization, Methodology, Writing - original draft, Writing – review and editing

### Author ORCIDs

Georgia Maria Sagia ⓘ https://orcid.org/0000-0002-7585-2035
George Diallinas ⓘ https://orcid.org/0000-0002-3426-726X
Sofia Dimou ⓘ https://orcid.org/0000-0002-2989-3426

### Decision letter and Author response

Decision letter https://doi.org/10.7554/eLife.103355.sa1
Author response https://doi.org/10.7554/eLife.103355.sa2

## Additional files

### Supplementary files
- MDAR checklist
- Supplementary file 1. Strains used in this study.
- Supplementary file 2. Annotation of proteins used in this study.
- Supplementary file 3. Primers used for cloning and gene construction.

### Data availability

Strains and plasmids are available upon request. The authors affirm that all data necessary for confirming the conclusions of the article are present within the article, figures, tables, and source data files.

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
