## [Editor Report]

This fundamental study advances our understanding of an unconventional route by which certain transmembrane proteins reach the plasma membrane of the fungus Aspergillus nidulans. By carefully examining two model plasma membrane proteins in parallel, the authors provide compelling evidence for their earlier proposal that sorting of the two proteins diverges upon ER export, with one protein following the standard secretory pathway while the other protein follows a Golgi-independent route. Even though a mechanistic understanding of the new pathway is not yet available, this work will be of interest to cell biologists who study membrane traffic.

---

## [Decision Letter]

[Editors' note: this paper was reviewed by Review Commons.]

---

## [Author Response]

Reviewer #1Sagia et al. present a manuscript using A. nidulans as model to study different transport routes of membrane proteins from the ER to the plasma membrane. They showed in earlier work that apparently at least two different transport routes exist, one involving the classical ER-ERES-ERGIC-Golgi route, one bypassing the Golgi. Unpolarized membrane proteins use the former, apically sorted membrane proteins the latter route. The study here confirms their earlier findings, uses a better model (co-expression of representatives for both routes in the same cell) and provides additional mechanistic insights about the roles of rabs, SNARES and other important proteins of the secretory pathway. The study is thoroughly done, figures are of high quality, data and methods well described and adequately replicated.

Thank you for your positive comments

I do have, however, a number of comments that could help to improve the manuscript.-I suggest using the term polarized or apical rather than polar. Polar alone to me refers more to physico-chemical properties like water-solubility.

Amended in most parts of the revised text.

-introduction and discussion: I don’t think the literature about unconventional secretion bypassing the Golgi is complete, for example studies about TMED10 like Zhang, M. et al. Cell 181, 637-652 e615 (2020) or Zhang et al. ELife 4 (2015) are missing, there might be others. Is UapA a leader-less cargo that could be inserted via TMED10 translocation?

Thank you for letting us know, we have missed these articles. More references on UPS are now added, including the Zhang et all publications. UapA, as all transporters, is a multispan transmembrane protein with no leader peptide. In fact, we have checked the role of p24 family proteins (homologous to TMED10) in UapA trafficking. The knock-out of key p24 proteins does not affect UapA sorting to the PM (please consider this as confidential unpublished results)

-Figure 1C. Can these intracellular structures be characterized in more detail?

As explained briefly to the handling editor above, and following the reviewer’s suggestion, we performed new experiments to better characterize the identity of the cargo-labeled fluorescent puncta. To do so, we used co-expression of a standard ERES marker, Sec16, in cells expressing either UapA or SynA, tagged with different fluorescent tags. More specifically, we constructed and analyzed strains co-expressing UapA-GFP/Sec16-mCherry or GFP-SynA/mCherry-Sec16 in the *sec31*^ts^ genetic background, which allows synchronization and better analysis of ER exit, as described in our text. The new findings appear as Figure 5C in the revised manuscript. Notice that *sec16-mCherry* introduced in the native *sec16* locus by standard knock-in reverse genetics of *A. nidulans* (see Materials and methods) does not affect Aspergillus growth or secretion. Experiments depicted in 5C show that both cargoes, UapA and SynA, co-localize significantly (PCC ≈ 0.6), with Sec16, suggesting that most of these puncta are indeed ERES structures. Given that the puncta marked with UapA or SynA are clearly distinct (see Figures 1C,2A, 3A, 5B), this new experiment strongly suggests that there are indeed two distinct ERES, one populated mostly by UapA and the other by SynA. Notice, as we already outline in our response to the editor above, a three-colored approach using Sec16-BFP (or Sec13-BFP) for showing directly the existence of these two populations of cargo-specific ERES in the same cell failed as the BFP signal was problematic for colocalization studies.

Where is the Golgi localized in A. nidulans, is it decentralized like in yeast?

Yes, as in *S. cerevisiae*, *A. nidulans* Golgi cisternae are individually scattered throughout the cytoplasm, also similarly to other filamentous fungi. Notice that in *A. nidulans* Golgi structures are moderately polarized (Pantazopoulou and Penalva 2009).

Is the UapA at the time points shown in Figure 1C in some sub-PM structures? To me the distribution at or near the PM is more punctate than in the steady state image shown in 1B

The punctuate appearance of PM transporters at the periphery of fungal cells is a common theme when these do not reach high, steady-state, levels of accumulation. In fact, several transporters mark specific subdomains of the PM, more evident before achieving their steady-state levels. For example, in yeast several amino acid and nucleobase transporters mark punctuate structures that colocalize with eisosomes markers (caveolin-like PM subdomains), while the proton pump ATPase Pma1 marks distinct punctuate domains. Similarly, UapA and other solute transporters mark punctuate structures before reaching their state-state accumulation in the PM. Figure 1C shows the de novo synthesis of cargoes after 100 min of transcription, while Figure 1B depicts the steady-state localization of UapA and SynA after 4h. In the latter case, the PM is ‘saturated’ with UapA molecules and thus the fluorescent signal of distinct puncta ‘fuses’, creating continuous fluorescent labeling. Notice also that in several cases, in our work, we have also performed UapA transport assays, which provide a direct tool to test and confirm the presence of UapA in the PM (see Figures 4D or 6C).

-Figure 3A. To me it looks like there is actually a lot of colocalization of UapA and SynA, especially at or near the PM, where there is quite some white, punctate staining. The green fluorescence is just much stronger, overlaying the violet. Can you show separate channels and explain?

We think the reviewer means Figure 2A, which compares UapA and SynA (Figure 3A compares UapA with Golgi markers). If so, we have quantitatively estimated and performed statistical analysis (PCC) which indicates that this, visually apparent colocalization, is not significant (right panel in Figure 2A). Notice also that we cannot totally exclude very minimal colocalization of UapA and SynA signals as both cargoes mark very proximal early secretory domains (i.e., ERES or ERGIC), especially in fungal cells. Anyhow, in the revised Figure 2 we also added a panel depicting separate channels, as the reviewer asks.

-Figure 3: In my opinion the statement that UapA "is probably sorted from an early secretory compartment, ultimately bypassing the need for Golgi maturation" is too strong at that point. You say for both UapA and SynA you don’t get significant colocalization with early Golgi/ERGIC marker, then you cannot conclude that one takes the conventional route via early-late Golgi and the other does not. What you can say is that UapA is apparently not going through late Golgi.

The reviewer is in principle correct. However, significant colocalization with the late Golgi marker, as SynA shows, strongly suggests that this cargo has passed via the early Golgi compartment. The fact we failed to detect significant colocalization of any cargo tested with early Golgi/ERGIC markers (e.g., SedV) is very probably due to very rapid passage of cargoes from these compartments, which conventional widefield or confocal microscopy cannot detect. To achieve this, ultra-fast fluorescent microcopy, as Lattice Light Sheet Microscopy (LLSM), should be used. In fact, we are currently initiating these studies, which will appear in the near future elsewhere.

-Figure 4C: UapA does not seem to accumulate in the ER in the Sec24 and 13 mutants but in punctate structures. This for me is unexpected, any explanations? Can you characterize that punctate staining?

This is an interesting observation. Notice that UapA is a large homodimeric protein (e.g., 28 transmembrane domains) that oligomerizes further upon translocation into the ER membrane. Repression of Sec24, and to a less extent of Sec13, leads to inability to exit the ER properly. Consequently, this will lead to UapA overaccumulation in the ER, which might in turn lead to ER stress and turnover, reflected in UapA aggregates. In line with this, we have previously shown that specific mutants of UapA unable to exit the ER are indeed degraded by selective autophagy (Evangelinos et al., 2016). In contrast to UapA, SynA partitions in the entire ER without forming aggregates when *sec24* or *sec13* are repressed. This might be due to the fact that is a single-pass, much smaller, membrane protein compared to UapA and one that is not known to form oligomers. Thus, its overaccumulation in the ER might not lead to aggregation, allowing it to diffuse laterally in the membrane of the ER. A note on this is included in the Figure legend of the revised manuscript.

-Figure 6D: You state that BFA "has only a very modest effect on UapA translocation to the PM". To me the PM (or very near PM) staining of UapA looks very different in the PFA treated cells, more uneven/punctate. Is there an explanation for that?

Our explanation is the following. When BFA is added, conventional secretion is blocked and Golgi collapses. We believe that this might have a moderate *indirect* effect also on cargoes bypassing the late Golgi/TGN, as UapA (i.e., lower levels of UapA present in the PM). This is based on the fact that UapA, in addition to conventional cargoes, requires the Q-SNARE complex SsoA/Sec9 to translocate to the PM. SsoA, being a membrane protein cargo itself, also needs to traffic to the PM. Interestingly, we have previously obtained evidence suggesting that SsoA traffics to the PM by both conventional and a Golgi-bypass routes (Dimou et al. 2020). Thus, UapA translocation to the PM might indeed be partially impeded or delayed due to repression of proteins, such as SsoA (and probably Sec9), needed for its final integration into the PM bilayer. Importantly, in line with an indirect effect of BFA on the levels of UapA localized in the PM, notice that, unlike SynA, UapA was never trapped in brefeldin bodies (i.e., Golgi aggregates).

Reviewer #1 (Significance):One strength of the study is the use of a model organism, A. nidulans, not cell cultures. Also, the use of both reporters, UapA and SynA, in the same cell is an advantage over previous studies using different lines and different promotors. Limitation of the study might be that it remains unclear to what extend the basic mechanism (UapA and SynA are transported to PM in different carrier and via different routes) can be generalized to other polarized (apically?) membrane proteins versus non-polarized membrane proteins in A. nidulans and whether a similar mechanism exists in other organisms. Some of the basic findings of the study are not new but were published by the same group. However, as the authors point out, the current study uses improved assays and extends their previous studies, advancing our understanding of the mechanistics of transport in the conventional secretory pathway and novel alternative routes. The study will be of interest for basic researchers in the trafficking field. My own expertise is transport through the secretory pathway in mammalian cells, many years ago more post-Golgi, now mostly ER-Golgi and ER itself.

We thank the reviewer for his positive comments.

Reviewer #2The idea that transmembrane proteins of the plasma membrane move from the ER to the Golgi and then to the cell surface is firmly entrenched, and the mechanisms and components of this secretory pathway have been extensively characterized. Secretory vesicles are often delivered from the Golgi to sites of polarized growth. This paper builds on previous work by the same group to provide evidence that in Aspergillus nidulans, some non-polarly localized plasma membrane proteins follow a very different pathway, which bypasses components of the conventional secretory machinery such as SNAREs that have been implicated in secretion as well as the exocyst. In particular, they systematically compare the trafficking of the SNARE SynA, which follows the conventional secretory pathway, with that of the purine transporter UapA, which apparently does not. The two proteins were co-expressed in the same cells using the same promoter. A variety of genetic and microscopy methods are used to support the conclusion that UapA reaches the plasma membrane by a route distinct from that followed by SynA.In my view, the authors present a convincing case. The individual experimental results are sometimes ambiguous, but the combined results favor the conclusion that UapA follows a novel pathway to the plasma membrane. I have only a few relatively minor comments.

Thank you for your positive comments

1. In the Introduction and elsewhere: to my knowledge, there is no clear evidence that AP-1-containing clathrin-coated vesicles carry cargoes from the Golgi to the plasma membrane. On the contrary, as recently reported by Robinson (https://pubmed.ncbi.nlm.nih.gov/38578286/), AP-1-containing vesicles likely mediate retrograde traffic in the late secretory pathway.

Thank you for this comment and the relative reference. We are aware that AP-1 is likely to also mediate retrograde traffic in the late secretory pathway or/and intra-Golgi recycling, as also reported by the group of Benjamin Glick. Thus, in the revised version we added a short comment on this plus relative references. Along this line, our previous work has shown that transcriptional repression of AP-1 arrests the polar localization of several apical markers in *A. nidulans* and we reported that this might be due to an effect on both anterograde and retrograde trafficking. Please see “Secretory Vesicle Polar Sorting, Endosome Recycling and Cytoskeleton Organization Require the AP-1 Complex in Aspergillus nidulans”. Martzoukou O, Diallinas G, Amillis S. Genetics. 2018 Aug;209(4):1121-1138. Overall, the fact that AP-1 was found absolutely dispensable for UapA trafficking, further strengthens our conclusion that UapA bypasses the Golgi.

2. In Figure 2, is there any known significance to the presence of UapA in "cytoplasmic oscillating thread structures decorated by pearl-like foci as well as a very faint vesicular/tubular network"?

At present we cannot answer this question. In order to understand what these structures represent and answer what is their role, we will need to employ super-resolution and ultra-fast microscopy and additional markers, which we envision to do. We suspect that they might be tubular networks, but this extends beyond the present work.

3. SynA is related to *S. cerevisiae* Snc1/2, which are known to be present in late Golgi compartments due to repeated rounds of endocytosis to the Golgi and exocytosis to the plasma membrane. The SynA shown here to colocalize with PH^osbp^ is probably present in a similar recycling loop rather than being en route to the plasma membrane for the first time. Therefore, the differential colocalization of UapA and SynA with PH^osbp^ does not by itself provide "strong evidence that the two cargoes studied traffic via different routes" as stated in the text but might instead indicate that only SynA undergoes frequent endocytosis. The text should be amended accordingly.

The reviewer is in principle correct. However, given that colocalization of SynA and PH^osbp^ occurred all over the cytoplasm of hyphae and not only at the apical region, and because we record colocalization of cargoes before their steady-state accumulation to the PM, thus at a stage where recycling must be minimal, the recorded colocalization should reflect anterograde transport rather than recycling. We added this reasoning it the revised text.

4. A missing piece of the story is a test of whether the puncta visualized for the two cargoes in Figure 5B are indeed distinct populations of COPII-containing ER exit sites. The relevant experiment would involve co-labeling of the cargoes together with a COPII marker. Three-color labeling would presumably be needed.

This point was also raised by reviewer 1 (and review 3) and thus performed new experiments to better characterize the identity of the cargo-labeled fluorescent puncta. To do so, we used co-expression of a standard ERES marker, Sec16, in cells expressing either UapA or SynA, tagged with different fluorescent tags. More specifically, we constructed and analyzed strains co-expressing UapA-GFP/Sec16-mCherry or GFP-SynA/Sec16-mCherry in the *sec31*^ts^ genetic background, which allows synchronization and better analysis of ER exit, as described in our text. The new findings appear as Figure 5C in the revised manuscript. Notice that *sec16-mCherry* introduced in the native *sec16* locus by standard knock-in reverse genetics of *A. nidulans* (see Materials and methods) does not affect Aspergillus growth or secretion. Experiments depicted in 5C show that both cargoes, UapA and SynA, co-localize significantly (PCC ≈ 0.6), with Sec16, suggesting that most of these puncta are indeed ERES structures. Given that the puncta marked with UapA or SynA are clearly distinct (see Figures 1C,2A, 3A, 5B), this new experiment strongly suggests that there are indeed two distinct ERES, one populated mostly by UapA and the other by SynA. Notice, as we already outline in our response to the editor above, a three-colored approach using Sec16-BFP (or Sec13-BFP) for showing directly the existence of these two populations of cargo-specific ERES in the same cell failed as the BFP signal was problematic for colocalization studies.

Reviewer #2 (Significance):This study provides compelling evidence that in the fungus Aspergillus nidulans, some transmembrane transporter proteins reach the plasma membrane by a pathway that bypasses much of the conventional machinery associated with the Golgi apparatus and secretory vesicles. Although previous publications pointed toward a similar conclusion, the present work tackles the problem in a more rigorous and systematic way. These findings are important for cell biologists who study membrane traffic, it remains to be determined how prevalent this type of non-canonical secretion might be in other organisms.

We thank the reviewer for his positive comments

Reviewer #3The manuscript by Sagia et al. compares the trafficking of a polarized (SynA) with a non-polarized (UapA) transmembrane protein. In agreement with previous work of the same lab, they find that UapA reaches the plasma membrane through a Golgi-bypass route, which they characterize to some extent. Overall, the data are of good quality and the story is interesting and timely. Understanding trafficking routes that bypass the Golgi is highly interesting. Nevertheless, there are several points of criticism that I have and below is a list where I combine major and minor points together:

Thank you for your positive comments

Major Comments:1- Is it possible that the polarized phenotype of SynA is caused by selective removal, i.e. SynA is delivered to the entire plasma membrane, but endocytosed rapidly from all areas except the tip of the hyphae. This would also result in a polarized distribution.

This is in principle possible, but here this is not the case. SynA is polarized due to rapid local endocytosis and immediate recycling at the subapical region, known as the subapical collar. Please see:

Taheri-Talesh N, Horio T, Araujo-Bazán L, Dou X, Espeso EA, Peñalva MA, Osmani SA, Oakley BR. *The tip growth apparatus of Aspergillus nidulans.* Mol Biol Cell. 2008 Apr;19(4):1439-49. doi: 10.1091/mbc.e07-05-0464.

Hernández-González M, Bravo-Plaza I, Pinar M, de Los Ríos V, Arst HN Jr, Peñalva MA. Endocytic recycling via the TGN underlies the polarized hyphal mode of life. PLoS Genet. 2018;14(4):e1007291. Published 2018 Apr 2. doi:10.1371/journal.pgen.1007291

This applies to all apical markers; they remain polarized by continuous local recycling after the diffuse laterally to the subapical collar.

2- The authors describe the distribution of SynA and UapA in cells deficient of various COPII/ERES proteins. However, these data are not shown, and it is not clear how they were quantified. It would be important to add quantitative data here.

Quantitative data are included in Figure 4C, displaying the percentages of cells with UapA either retained in the ER or reaching the PM for each background deficient in a COPII protein. Repression of SarA and Sec31 resulted in UapA retention in the ER in all analyzed cells (100%). However, repression of Sec12, Sec24, or Sec13 had a differential effect across the cell population, with UapA reaching the PM in some cells, while remaining trapped in the ER in others. To quantify these data and determine which cargo localization pattern prevails, we measured the number of cells in each category and represented them as percentages. A similar approach was used to examine the role of Golgi proteins in the trafficking of UapA and SynA (Figure 6).

3- on page 8, the authors discuss the discrepancy regarding the role of Sec13. They offer as an explanation that the previous studies have been performed in strains that separately expressed the two cargoes. However, I am unable to see why and how this would be a valid explanation.

Given that Sec13 has a variable/partial effect on UapA, we have previously been biased towards images that showed an effect on localization, as expected, and considered that the lack of an effect might have been due to inefficient repression in a fraction of cells. In our new system, we were able to directly compare UapA to SynA and find out that while SynA was always affected under our conditions, the effect of UapA was still variable. Thus, the partial effect of Sec13 on UapA is physiologically valid and not a matter of insufficient repression in a fraction of cells. This shows the importance of our new improved system where we follow the synchronous expression of two cargoes in the same cells.

4- Why is the effect of Sec24 depletion so much stronger than of Sec12 depletion? Sec12 is the GEF for SarA, without which Sec24 should not be recruited to ERES. The explanation that low amounts of Sec12 are still present and sufficient to carry out the role of this protein. What is the evidence for that?

Sec24 is the principal receptor of cargoes responsible for their recruitment to ERES. Sec12 is the catalytic effector for SarA required for the initiation of COPII vesicle formation. The question of the reviewer is thus logical.

However, Sec12 is indeed present at extremely very low levels when expressed from its native promoter under the condition of our experiment (minimal media). This is supported by our recent proteomic analysis, performed under similar conditions, which failed to detect the Sec12 protein, unlike all other COPII components (see Dimou et al., 2021, doi; 10.3390/jof7070560), but also by cellular studies of the group of M.A. Peñalva, who failed to detect Sec12 tagged with GFP (Bravo-Plaza et al., 2019, doi: 10.1016/j.bbamcr.2019.118551). Additionally, in yeast, immune detection of Sec12 has been possible only in cells harboring sec12 on a multicopy plasmid, suggesting its low abundance in wild-type cells (Nakano et al., 1988, doi:10.1083/jcb.107.3.851).

Given that repression of *sec12* transcription via the *thiAp* promoter still allows 68% of cells to secrete normally both SynA and UapA, while 32% of cells are blocked in the trafficking of both cargoes, suggests that in most cells either SarA can catalyze the exchange of GDP for GTP without Sec12, maybe through a cryptic guanine nucleotide exchange factor (GEF), or that very small amounts of Sec12 remaining after repression are sufficient for significant SarA activation. Whichever scenario is true, Sec12, similarly to SarA, is not critical for distinguishing Golgi-dependent from Golgi-independent routes, as both cargoes are affected similarly. In the revised text we added a not on this issue.

5- In Figure 5, it would help readers who are not so familiar with Aspergillus organelle morphology to explain the figure a bit better. This might appear trivial for experts, but anyone from outside this field is slightly lost.

In the revised manuscript we added a figure panel depicting a schematic representation of *A. nidulans* key secretory compartments.

6- The authors write that not seeing UapA in Golgi membranes is evidence that it does not pass through this organelle. However, when they write that SynA is never seen in cis-Golgi elements, they do not conclude that SynA bypasses the cis-Golgi.

The fact that SynA, unlike UapA, colocalized significantly with late-Golgi/TGN and follows conventional secretion in general, strongly suggests that SynA also passes from the early-Golgi. Cargo traffic through the Golgi is mediated by cisternal maturation, where an individual cisterna gradually changes its nature from an earlier to a later one, while the cargo remains inside. UapA, unlike SynA, never colocalized with any Golgi marker used and was not affected by BFA. We agree with the reviewer that we did not have direct proof for passage of UapA or SynA from the early-Golgi in the wt background, which allows for the alternative, but rather unlikely hypothesis, that none of the two cargos is sorted to the early Golgi and that SynA traffics directly to late-Golgi/TGN. Our inability to detect sorting of any cargo to the early-Golgi is seemingly due to ultra-fast passage of cargoes from very early secretory compartments, such as ERGIC/early-Golgi. In fact, we have obtained evidence of this using Lattice Light Sheet microscopy (results in progress, to appear elsewhere).

7- Figure 5C: the authors claim that the CopA and ArfA affects trafficking of UapA and SynA from ER to plasma membrane and assign CopA and ArfA as regulators for anterograde trafficking. I think this interpretation is not justified by the data. Depletion of CopA and ArfA will affect the Golgi apparatus in structure and function. The more straight-forward interpretation is that repression of the COPI machinery results in a defect in Golgi exit and therefore retention in pre-Golgi compartments (including the ER and maybe the ERGIC should it exist in Aspergillus). The same is true for BFA treatment where there are also negative effects on ER export, which are rather indirect consequences of alterations of Golgi function and integrity. Likewise, the interpretation of the papers by Weigel et al. and Shomron et al. is not correct. It is more likely that COPI is recruited to the growing ERES-derived tubule (or ERGIC) to recycle proteins back to the ER. This is not necessarily a proof that COPI regulates anterograde trafficking

This is a highly debatable issue which our work cannot address. However, we amended the text accordingly.

8- Figure 6: The images look like in Figure 5, yet here you don't call them ER-associated.

The two images are not alike. In Figure 5 upon activation of Sec31 (permissive temperature) we detect mostly punctual structures resembling ERES, whereas at the nonpermissive temperature we detect a membranous network typical of the ER. Upon repression of CopA we also detect punctual structures similar to ERES. In Figure 6, we mostly detect an effect on SynA. Repression of early secretory steps (SedV, GeaA) lead to collapse of SynA in the entire ER network. Repression at later stages of Golgi maturation and post-Golgi secretion (RabO, HypB, RabE, AP-1) lead to the appearance of punctual structures, most probably Golgi aggregates.

9- Figure 6D: How long was the BFA treatment. I am surprised that the pool of SynA preexisting at the plasma membrane seems to also be sensitive to BFA.

Cells were grown overnight under repressed conditions for both UapA and SynA. After 12-14h cells were shifted to derepressed conditions using fructose as carbon source. BFA was added after 90min of cargo derepression, while both cargoes were still in cytoplasmic structures so there was not preexisting SynA or UapA at the PM (see also Figure 1C). Subcellular localization of both cargoes was studied for 60min after BFA treatment.

10- This might be beyond the scope of this study, but as far as I know UapA is not N-glycosylated. Would the introduction of an N-glycosylation site shift it towards the Golgi-based route?

Thank you for this suggestion. We have performed this experiment, adding a glycosylation site on UapA, based on the glycosylation sites found in tis mammalians homologues. We did not detect any effect on UapA trafficking route or its activity. As the reviewer recognizes this goes beyond the scope of this study and thus, we did not include it the manuscript. Differential cargo glycosylation is however an important issue to be studied systemically in respect to different trafficking routes, and we envision to investigate it systematically.

Minor Comments1- This might be just a personal preference, but I think that the term polar is misleading, because it implies something about the polarity of the amino acids. I think "polarized" might be the more common term. Anyway, this is just a minor point and just a suggestion from my side.

Amended in the revised text.

2- The paper by the Saraste lab should be mentioned and discussed (PMID: 16421253), which I think is very relevant to the current story.

We thank the reviewer for pointing out this important publication. In that case, the Rab1 GTPase defined a pathway connecting a pre-Golgi intermediate compartment with the PM in mammalians nerve cells. Thus, the Saraste lab publication is indeed along the lines of findings supporting that Golgi-independent unconventional cargo trafficking routes initiate at very early secretory compartments. Notice, however, that RabO, the *A. nidulans* homologue of Rab1, which in their case was essential for direct cargo sorting from the ERES/ERGIC to the PM, in or system, was dispensable for Golgi bypass. The Saraste lab article is now mentioned and discussed.

3- Having worked with ERES for over two decades, I find it strange to see it written ERes. I see no reason why ER exit sites in Aspergillus should be abbreviated differently from all other types of cells (yeast, *Drosophila*, worms, mammals). I think that the entire acronym should be capitalized.

Amended in the text

4- When discussing the data about the partial effect of Sec13, it would be good to refer to a previous paper by the Stephens lab that showed that silencing Sec13/31 results in a defect in trafficking of collagen, but not of VSVG (PMID: 18713835).

We thank the reviewer for also pointing out the publication of the Stephens lab, now mentioned in the revised text. Noticeably, in that case silencing of both Sec13 and Sec31 has no effect on the trafficking of specific cargoes, whereas in our case Sec31 is still absolutely needed for both conventional and Golgi-independent secretion of SynA and UapA, respectively.

Reviewer #3 (Significance):Overall, the data are of good quality and the story is interesting and timely. Understanding trafficking routes that bypass the Golgi is highly interesting. The main weakness is the lack of mechanistic understanding of the Golgi-bypass pathway. In addition, the study is limited to two proteins as representatives of polarized vs. non-polarized proteins. The main target audience for this paper are scientists working in the area of secretion and trafficking in the secretory pathway.

We thank the reviewer for his positive comments.

We are aware that the mechanistic details of Golgi bypass are missing and this is our next goal, dissecting those via various approaches genetic and biochemical approaches and employment of super resolution and ultra-fast microscopy.

Reviewer #4In this study, Sagia et al. investigate the trafficking of different secretory cargo in Aspergillus nidulans under conditions that repress expression of transport factors or block stages in membrane trafficking. The primary approach is to conduct dual live-cell imaging of GFP-tagged UapA (plasma membrane localized purine transporter) and SynA (plasma membrane R-SNARE) after their simultaneous derepression to monitor trafficking routes. In germlings, both secretory proteins are detected in non-overlapping intracellular compartments and puncta after 60-90 min of derepression. After 4-6 hrs, SynA localizes to hyphal tips whereas UapA localizes to non-polar regions of the PM. Colocalization studies do not show UapA overlap with Golgi markers (SedV, PH-OSBP) during its biogenesis whereas SynA displays significant co-localization. Repression of COPII and COPI components generally block transport of both cargos to the PM and cause accumulation in ER compartments, although there are some differential effects on UapA and SynA localization. Finally, repression of other transport factors (ER-Golgi SNAREs, Golgi transport factors, and exocytic machinery) had differential effects on UapA and SynA localization over time with UapA reaching the plasma membrane in many instances and SynA accumulating in intracellular compartments.Based on these observations, the authors conclude that UapA and SynA follow distinct trafficking routes to the plasma membrane where SynA uses a canonical SNARE-dependent secretory pathway route and UapA follows a non-canonical route that may bypass Golgi compartments. The study is extensive and supports the model that biogenesis of SynA and UapA follow distinct processes. However, there are some complexities that may limit interpretation. First, the cargo studied are targeted to the ER differently. UapA is a multispanning transmembrane protein that is likely dependent on the Sec61 translocon for co-translational membrane insertion and will involve ER chaperones and quality control machinery for its biogenesis. SynA will depend on the tail-anchored machinery (GET/TRC pathway) for insertion into the ER and is processed by cytosolic factors/chaperones. Therefore, the sites of ER insertion and the rates of biogenesis of these cargoes will be different. In addition, the repression of trafficking machinery used in this study appears to be variable and may exert partial blocks on intracellular transport stages. Regardless, the study clearly documents that SynA and UapA follow distinct biogenesis and transport processes when co-expressed in cells under experimentally controlled conditions.

Thank you for your positive comments.

To our knowledge there is no evidence suggesting that SynA translocates via a tail-anchored machinery (GET/TRC pathway) and not through the translocase. Despite this, we agree with the reviewer that translocation to the ER, as well as exit from it, might be cargo-dependent, especially when it concerns proteins with very different size, structures and oligomerization. Thus, the rate of biogenesis of UapA and SynA is probably quite different. However, this still does not dismiss our basic conclusion that the two cargoes follow distinct routes to traffic to the PM. The ‘problem’ of variable transcriptional repression of some trafficking-related proteins is solved by comparing the relative effect of the two cargoes in the same cells, and this is in fact the advantage of our new system. Importantly, notice that we took care to use conditions of repression where SynA trafficking by the conventional path was totally abolished and compared it to UapA.

We have performed a critical experiment showing colocalization with Sec16, confirming partition of the two trafficking routes at early secretory compartment. This, together with the differential dependence of the two cargos on COPII components proves the existence of two kinds of early carriers. This is an important advancement compared to our previous publications for dissecting the full mechanism of Golgi bypass.

Our studies did not only include co-localization experiments. Our work additionally and primarily includes a series of experiments performed in strains KO for conventional secretion, where UapA reaches the PM while SynA does not. Differential dynamics of insertion of the two in the ER cannot explain the steady state accumulation of UapA, but not of SynA, in the PM when conventional secretion is blocked.

1. It was not clear if the translation, ER insertion and folding of UapA and SynA are fully synchronous. Is it possible that the rate of UapA synthesis and transport to the plasma membrane is substantially faster than for SynA? The imposition of transport blocks could trap SynA and not UapA if this cargo was at later transport stages.

As already discussed above translation, ER insertion and folding of UapA and SynA might indeed by different. This might somehow affect the trafficking path followed, but this issue is beyond the scope of this work. Notice, however, that the transcription of both cargoes is kept fully repressed during establishment of repression of secretion. Only when repression and blocking of secretion is established (12-14 h germination), as verified by Western blot analysis, we derepress the transcription of UapA and SynA, expressed from the same promoter, and follow their dynamic subcellular localization. Hence, this system ensures that both cargoes start from the earliest transport stage, the ER, upon imposition of transport blocks.

2. In repressing transport factors (e.g., SarA, Sec12, Sec24, Sec13, SedV, RabE), it is clear that under thiamine repressing conditions these cells do not grow or have greatly reduced growth rates. However, it was not clear if proteins are depleted to the same extent in cells after repression for 12-14 hr or 16-22 hr. as mentioned in the methods. Indeed, in some cases depleted cells display different cargo localization patterns, for example 67% of cells show normal localization of UapA and SynA after sec12 repression and 33% show ER accumulation of both cargoes. There is differential localization of UapA and SynA in many cases where transport factors are repressed, but this could be due to partial inhibition and not complete blocks. It would be helpful to clearly indicate the time points and conditions in each of the figure legends as in points 3-5 below.

In the revised manuscript we did our best to clearly indicate the time points and conditions in each of the figure legends. Differential localization of UapA and SynA in many cases where trafficking factors are repressed is indeed an interesting outcome. Inefficient repression was dismissed based on the lack of colony growth (see relative growth tests of SarA, Sec24, Sec13, Sec31, SedV, GeaA, RabO, RabE, Ykt6, Sft1, SsoA and Sec9), but also by western blots (e.g., Sec24, Sec13, Sec31 or Sec9 shown in the present manuscript, or other trafficking proteins studied previously. Martzoukou et al., 2018; Dimou et al., 2020). Repression of Sec12 and HypB, and to lower degree AP-1, allowed formation of small and/or compact colonies, but even in these cases relative protein levels could not be detected in western blots, guaranteeing efficient repression.

3. In Figure 4A immunoblot, HA-tagged proteins are not detected after thiamine repression. Please state the time of thiamine repression used before protein extraction and blot. Is this for the same length of time as for cells shown in panel 4C? It would also be helpful to state the time of cargo derepression before capturing images in 4C. The methods section mentions 12-14 hr or 16-22 hr of growth, presumably with thiamine in the culture, and then 1-8 hr or 60 min to 4 hr of cargo derepression before imaging. Please specify.

The time of thiamine repression before protein extraction was 16-18h. The same repression time was used for experiments shown in Figures 4C and 6C (ER/COPII and Golgi/post-Golgi repression respectively). More specifically, for microscopy experiments cells were grown in the presence of glucose and thiamine for 12-14h (repressed UapA/SynA and *thiA_p_* expressed gene). After this time, cells were shifted to fructose and thiamine for 4h (derepression of UapA/SynA and repression of *thiA_p_* expressed gene). In both cases (protein extraction and microscopy experiments) the total time of thiamine repression was 16-18h.

4. For the thiA-copA and thiA-arfA repression experiments (Figure 5C), the methods section states that thiamine was not added ab initio in the culture, but after an 8 h time window without thiamine at the start of spore incubation. This is interpreted to mean that repression was for a shorter period to time than the 12-14 hr overnight growth. However, the figure legend states that de novo synthesis of cargos takes place after full repression of CopA and ArfA is achieved (>16 hr). Please clarify.

We think that the review was confused with repression of cargo synthesis (via alcAp+glucose) versus repression of trafficking proteins (via *thiAp*+thiamine). Please see Materials and methods. We clarify our protocol also here:

For the *thiAp-copA* and *thiAp-arfA* repression experiments addition of thiamine *ab initio* in the culture leads to total arrest of spore germination and germling formation. Thus, we added an 8-hour time window without thiamine to allow conidiospores to germinate until the stage of young germlings, under conditions where cargo expression via the *alcAp* was repressed by glucose. Subsequently, thiamine was added in the media (16-18 h) to repress CopA and ArfA, while cargo expression remained glucose-repressed. The transcriptional repression of the cargoes UapA and SynA was maintained for a longer period (24-26 h) compared to other repression experiments, but longer times of repression of cargoes do not make any difference, as full repression is achieved already at 12 h. de novo cargo trafficking was followed next day by eliciting depression, via a shift to fructose media, while still maintaining thiamine to repress CopA or ArfA.

5. In Figure 6D, BFA treatment is shown to trap SynA in Golgi aggregates while UapA still reaches the plasma membrane. Please state the time of BFA treatment before collecting these images. Do longer treatments with BFA before cargo derepression cause accumulation of UapA in intracellular compartments?

As mentioned above (response to Reviewer’s #3 comment 9) cells were grown overnight under repressed conditions for both UapA and SynA. After 12-14h cells were shifted to derepressed conditions using fructose as carbon source. BFA was added after 90min of cargo derepression, while both cargoes were still in cytoplasmic structures so there was not preexisting SynA or UapA at the PM (see also Figure 1C). We have not noticed any different effect on UapA trafficking after a max of 1h of BFA treatment.

6. A minor point, but on page 21 the methods state that "cells were shifted down to the permissive temperature (25 C), to restore the secretory block…". Suggest changing to "to reverse the secretory block…"

Modified accordingly

Reviewer #4 (Significance):This manuscript nicely builds on a developing line of investigation in the Aspergillus nidulans model that specific plasma membrane proteins are efficiently delivered to the cell surface in a pathway that is distinct from the canonical secretory pathway. Previous work from this lab has suggested that a subpopulation of COPII carriers can bypass the Golgi for delivery of specific cargo to the plasma membrane. The current study uses dual expression of UapA-GFP and mCherry-SynA to provide further support for this model. Molecular definition of a direct ER to PM transport pathway for secretory cargo would be a significant advance to a broad audience. This study provides additional depth and support that such a pathway exists but does not define how COPII vesicles or related intermediates are transported to the PM.

Again, thank you for your positive comments.